# Observation of the photonic Hall effect and photonic magnetoresistance in random lasers

Wenyu Du[1,2,3], Lei Hu[1,2], Jiangying Xia[1,2], Lin Zhang[4], Siqi Li[1,2,3], Yan Kuai[1,2], Zhigang Cao[1,2,3], Feng Xu[2,3], Yu Liu[1,2,3], Kaiming Zhou[4], Kang Xie[5], Benli Yu[1,2,3], Ernesto P. Raposo[6], Anderson S. L. Gomes[7] & Zhijia Hu[1,2,3] ✉

Modulation of scattering in random lasers (RLs) by magnetic fields has attracted much attention due to its rich physical insights. We fabricate magnetic gain polymer optical fiber to generate RLs. From macroscopic experimental phenomena, with the increase of the magnetic field strength, the magnetic transverse photocurrent exists in disordered multiple scattering of RLs and the emission intensity of RLs decreases, which is the experimental observation of photonic Hall effect (PHE) and photonic magnetoresistance (PMR) in RLs. At the microscopic level, based on the field dependence theory of magnetic disorder in scattered nanoparticles and the replica symmetry breaking theory, the magnetic-induced transverse diffusion of photons reduces the scattering disorder, and then decreases the intensity fluctuation disorder of RLs. Our work establishes a connection between the above two effects and RLs, visualizes the influence of magnetic field on RL scattering at the microscopic level, which is crucial for the design of RLs.

When light propagates in a scattering medium subjected to a transverse magnetic field, it is deflected in the direction perpendicular to the incident beam and magnetic field, a phenomenon known as the photonic Hall effect (PHE)[1,2]. This effect has strong phenomenological similarities to the electronic Hall effect, in which the Lorentz force causes the deflection of the electron trajectory. The origin of PHE is a magnetically induced change in the optical scattering properties of a single particle in a disordered medium. It is worth noting that the PHE should be distinguished from two other effects with similar names, the photon spin Hall effect (PSHE) and the light Hall effect. For the spin Hall effect, the symmetry of the electromagnetism allows the spin of electrons to be deflected by the electric field. In an analogous way, PSHE refers to the spin property of magnetic components in photons. After a beam passes through an optical interface or a heterogeneous medium, photons with opposite spin angular momentum undergo lateral spin separation, which is manifested as spin-dependent splitting[3,4]. PSHE does not require the presence of an applied magnetic field[5]. In the PHE, the external magnetic field induces the deflection of scattered light and generates magnetic transverse photocurrent. The Hall effect of light is a description of the wave property of light, in which the interaction between the orbital angular momentum and the spin angular momentum leads to the displacement of the wave packet motion perpendicular to the dielectric constant gradient[6]. While these three effects are likely related, we remark that this work focuses on the experimental observation of PHE in random lasers (RLs) systems. We also mention that B. A. van Tiggelen's team has found PHE in numerous systems, including but not limited to magnetic scattering nanoparticles[2], atomic hydrogen[7], and absorbing media[8].

[1]Information Materials and Intelligent Sensing Laboratory of Anhui Province, Anhui University, Hefei 230601, China. [2]Key Laboratory of Opto-Electronic Information Acquisition and Manipulation of Ministry of Education, Anhui University, Hefei 230601, China. [3]School of Physics and Opto-electronics Engineering, Anhui University, Hefei 230601, China. [4]Aston Institute of Photonic Technologies, Aston University, Birmingham B4 7ET, UK. [5]School of Opto-Electronic Engineering, Zaozhuang University, Zaozhuang 277160 Shandong, China. [6]Laboratório de Física Teórica e Computacional, Departamento de Física, Universidade Federal de Pernambuco, 50670-901 Recife, PE, Brazil. [7]Departamento de Física, Universidade Federal de Pernambuco, 50670-901 Recife, PE, Brazil. ✉e-mail: zhijiahu@ahu.edu.cn

The generation mechanism of the PHE also produces photonic magnetoresistance (PMR)[9], which is similar to electron magnetoresistance. The transmission of multiply scattered light through a random medium is related to the applied magnetic field, and the relative transmittance decreases with the increasing magnetic field. B. A. van Tiggelen and coauthors reported the observation of PMR in scattering media[9]. The transmittance perpendicular to the magnetic field direction is proportional to the squared magnetic field strength. K. J. Chao et al. have found the anisotropic PMR in dense Co particle ensembles[10] and demonstrate the isotropic PMR of far-infrared transmission in ferromagnetic particles[11]. Both PHE and PMR show the modulation of multiple scattering by the magnetic field, which is crucial for RLs, where multiple scattering plays a decisive role[8].

RLs differ greatly from traditional lasers because they lack the normal resonant cavity structure[12]. RLs are a prime example of a complex physical system and a key focal point for multidisciplinary research. RLs have attracted extensive research interest from researchers in materials science[13], physics[14], sensing[15], biomedicine[16], optical communication[17], and other fields. From an application standpoint, the inherent randomness of RLs stands out as a rare embodiment of true randomness[18], garnering significant interest in fields such as encryption[17] and time-domain ghost imaging[12]. Building on fundamental principles of physics and mathematics, researchers endeavor to discern underlying order within the phenomenon and mechanism of RLs disorder while seeking to regulate this disorder through systematic approaches[19]. As a result, more and more ordered regularities are found in RLs, while conventional lasers are introduced and benefiting from disordered elements. The effect of magnetic fields on RLs was explored theoretically by Pinheiro[20], who highlighted the correlation between the quality factor $Q$ of RLs and the external magnetic field. Pinheiro proposed a category of RLs utilizing magneto-optical scatters. Al-Samak et al. examined the impact of the dynamic characteristics of $Fe_3O_4$ magnetic nanofluid dispersions with different concentrations on the transmission of Gaussian laser beams through them, including the role of the magnetic field intensity and the response time. Their results indicate that the scattering mean free path $l_s$ is greatly modified by varying the magnetic field strength[21].

These advanced reports raised some significant but still unveiled questions about the existence of PHE and PMR in RLs. To our knowledge, these two phenomena have never been reported in RL systems so far. To explore whether PHE and PMR exist in RL, we need to find a suitable RL system and a suitable representation method to link the macroscopic experimental results brought by PHE and PMR with the microscopic regulation of disordered scattering by magnetic field.

It is possible to obtain RLs from various gain-scattering systems, such as polymers[22], liquid crystals[23], quantum dots[24], semiconductors[25], etc. Among these systems, the two-dimensional waveguide confinement of the optical fiber can form a "total reflection-scattering-total reflection" scattering system, which reduces the threshold of RL and restricts the RL emission to a cone-shaped orientation[26]. In addition, the polymeric materials are also quite diverse and inclusive, they can be doped with the appropriate structure and present interesting properties, which expands the potential applications for RL modulation[27]. Therefore, doping the magnetic scattering medium and gain material in the core of a polymer optical fiber (POF) may constitute a suitable RL configuration for exploring PHE and PMR effects.

For the second requirement, the unique structure of the RL provides superior characteristics. As a result of the disorder of multiple scattering, the intensity and position of multiple principal peaks in the RL also display random fluctuations within a certain range[12], which is the microscopic scattering disorder in the macroscopic manifestation. Angelani et al. investigated the nonlinear dynamics of multimode lasers using statistical physics of disordered systems, demonstrating that light exhibits glassy behavior as it propagates through a random nonlinear medium and predicting the replica symmetry breaking (RSB) phase transition as a function of the pump intensity[28–30]. Spin-glass theory is one of the frontier research paradigms in complex physics, which describes neural networks and biological systems, stochastic photonics, and many other research areas. According to this theory, the same system can reach different states under the same conditions. This behavior is similar to the theoretical representation of the spin-glass, ferromagnetic, and paramagnetic phases, as well as the transitions between them. The glassy structure of silica, for example, is amorphous, exhibiting a disordered nature. In contrast, the quartz structure of silica demonstrates a crystalline arrangement, which is characterized by regular crystal symmetry. Analogously, the spin-glass and paramagnetic states are observed in magnetic spins. The initial experimental demonstration of RSB with Rayleigh scattering in disordered media was conducted by Ghofraniha et al.[31] through the incorporation of oligomeric dyes into amorphous solids[31]. Subsequently, Gomes et al. observed RSB in a one-dimensional continuous erbium-doped fiber RL[32] and in a regime of RL with $YBO_3^-$ doped $Nd^{3+}$ crystal powder[33]. Hu's team fabricated POF RLs using an electrostatic spinning technique and observed distinct paramagnetic and spin-glass phases at different pump energies[34,35]. The phase transition process of RSB is also well represented in the modulation process of RLs. Hu et al. presented the experimental proof for the tunable RSB in RLs by realizing the temperature and structurally controllable RSB in POF RLs[36]. This indicates that the structure-tunable RSB in a POF RL could be caused by variations in the degree of disorder within the POFs. The evaluation of scattering disorder in RLs is an imperative issue to be solved. The above works inspired us to use the RSB phenomenon in spin-glass theory to characterize the modulation degree of the magnetic field on RL disorder multiple scattering and, in turn, to explore how PHE and PMR affect the RLs emission.

In this work, we dope $Fe_3O_4@SiO_2$ magnetic nanoparticles with core-shell structure into the fiber core of POF to generate RL, and experimentally observe PHE and PMR in RL. For PHE, the applied magnetic field causes the RL to generate a magnetic transverse photocurrent. For PMR, we observe that the intensity of RL decreases with increasing magnetic field strength. To explore the effect of PHE and PMR on RL disordered multiple scattering, we first determine that the scattering disorder of RLs decreases with increasing magnetic field based on the field-dependent theory of magnetic disorder in nanoparticles. Then RSB in the spin-glass theory is applied to analyze the role of the magnetic field in regulating the degree of random lasing scattering disorder. Our findings suggest that the scattering disorder of the RL decreases as the magnetic field increases due to the effect of the magnetic dipole spin and scattering cross-section. This study highlights a regulation rule in disordered RL multiple scattering and provides a more in-depth analysis of the RL RSB theory from the perspective of the disorder.

## Results

### Theoretical framework

As shown in Fig. 1, the pump laser is coupled into one end of the magnetic gain polymer optical fiber (MGPOF), the generated RL and the residual pump laser exit from the other end of MGPOF, which are oriented along the x-axis. The direction of the applied magnetic field is along the z-axis. The RL is formed by the disordered multiple scattering of photons in the gain medium. From the macroscopic experimental phenomenon, the applied magnetic field causes the RL to produce magnetic transverse photocurrent $J$, which proves the PHE in the RL system. Moreover, the intensity of the RL emission is weakened by the applied magnetic field, which proves the PMR in the RL system. From the microscopic disordered multiple scattering during RL generation, we note that contrary to the commonly assumed view that nanoparticles have a constant global magnetic moment, in fact, the applied magnetic field can significantly increase the magnetic moment

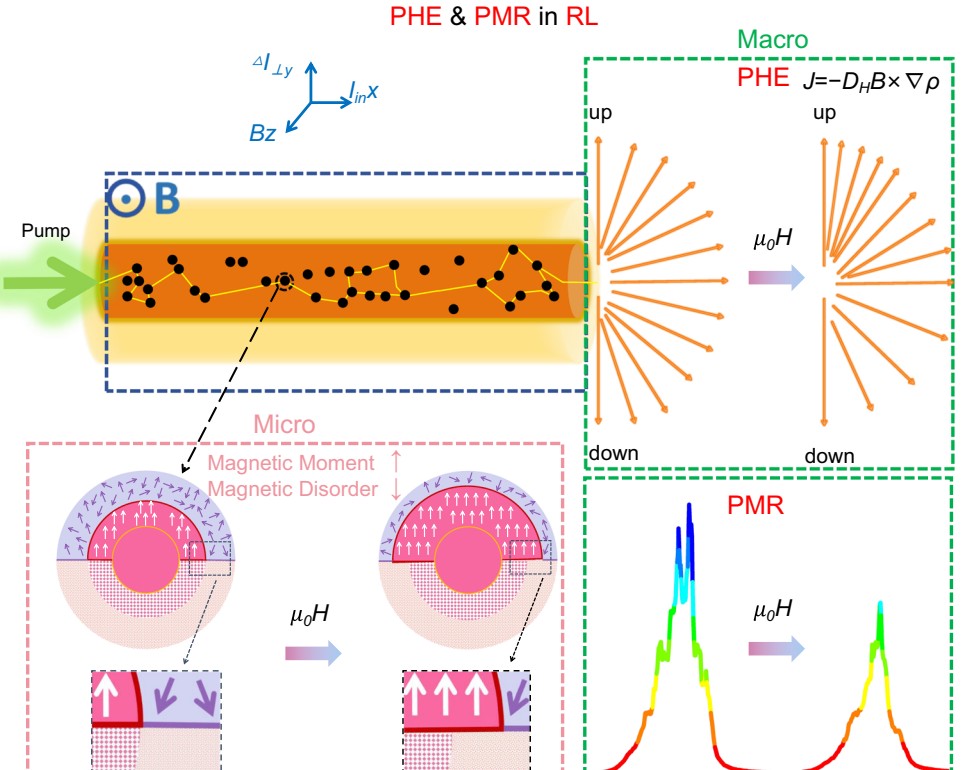

**Fig. 1 | PHE and PMR in RL system. The top left is the MGPOF with Fe₃O₄@SiO₂ magnetic nanoparticles randomly distributed in the core.** After the pump light is coupled into the MGPOF, RL is generated by disordered multiple scattering of photons in the gain medium. The MGPOF is completely in the applied magnetic field. The two green boxes show the macroscopic phenomena resulting from the magnetic regulation of RL. The green box in the upper right corner shows the magnetic transverse photocurrent generated by RL under the applied magnetic field, which corresponds to the PHE in RL system. The green box in the lower right corner shows that the emission intensity of RL is weakened under the applied magnetic field, which corresponds to the PMR phenomenon in RL system. The pink box in the bottom right corner shows the microscopic modulation effect of the applied magnetic field on RL. Under the action of the external magnetic field, the Fe₃O₄@SiO₂ magnetic scattering nanoparticles overcome the structural disorder on the surface, and the radius of the ordered magnetic moment increases, which weakens the RL scattering disorder and then weakens the RL intensity fluctuation disorder.

of ferrite nanoparticles. The magnetization volume of nanoparticles is closely related to the surface disorder of their structure, and when a magnetic field is applied, the disordered surface spin is gradually polarized, resulting in an increase of more than 20% in the magnetic volume[37]. In ref. [37], the magnetic scattering amplitude of small angle neutron scattering (SANS) is used to determine the morphology of magnetic nanoparticles. It is found that the magnetic core radius $r_{mag}$ increases with the increase of the external magnetic field strength. The variation of the severe magnetic scattering amplitude with the applied magnetic field is simulated based on the micromagnetic theory. The simulation results show that the fluctuation of magnetic parameters, that is, the contribution of magnetocrystalline anisotropy and magnetostriction, is the most likely source of the variation of magnetic radius with magnetic field. The magnetic volume and the corresponding magnetic field energy increase with the increase of the applied magnetic field strength are obtained by calculating the Zeeman free energy. The core-shell Fe₃O₄@SiO₂ NPs used in this work and the CoFe₂O₄ NPs sphere in [37] surrounded by the oleic acid ligand layer behave similarly. That is, the size of the magnetized core increases with the applied magnetic field because they have similar high surface-to-volume ratios, similar surface atom distribution, and similar magnetic resonance lines in the range 0–800 mT[38–40].

As shown in the pink box of Fig. 1. The upper semicircle represents the structural morphology, and the lower semicircle represents the magnetic morphology. Prior to the application of a magnetic field, the structural and magnetic morphology of nanoparticles are equal in size, as shown in the circle on the left, whereas the initially disordered surface spin is gradually polarized in the applied magnetic field,

causing the magnetic radius to increase beyond the structurally disordered surface region as shown in the circle on the right. The deep pink squares show the structurally coherent grain of the Fe₃O₄ core, and the light pink dots show the structural disorder of the Fe₃O₄ surface region. Note that the SiO₂ shell is not shown in the morphology of ferrite NPs. The silica shell on each NPs has its amorphous nature. The white arrows represent the collinear magnetic dipole spin in the magnetic core, and the purple arrows represent the spin disorder of Fe₃O₄ surface region. The overall magnetic moment of the core-shell structure Fe₃O₄@SiO₂ magnetic NPs increases with the magnetic field. Following the Faraday effect, the magnetic field causes the rotation of the scattering polarization plane, which is linearly proportional to the component of the magnetic field toward the direction of light wave propagation. An increase in the applied magnetic field leads to an enhanced effect on the orientation of scattered light, causing an increase in the radius of the ordered magnetic dipole moment. This results in a weakened scattering disorder of RL, which ultimately reduces the RL intensity fluctuations.

In disordered multiple scattering, the anisotropy of the scattering cross-section is determined by the "anisotropy factor" $<cos\theta>$, which is the average of $cos\theta$ divided by the phase function[41]. $\theta$ is the angle of scattering deflection. In magnetic multiple scattering, $<cos\theta>$ can separate "up" scattering from "down" scattering. If the magnetic field is perpendicular to the incoming and outgoing wave vectors, there is a difference between the upward and downward flux along the magneto-transverse direction $\mathbf{k} \times \mathbf{B}$. The magnetic anisotropy $\eta(\eta \neq 0)$ of a single scatterer in magnetic multiple scattering can be precisely quantified as the normalized difference between the total upward flux

and the total downward flux as:

$$\eta \equiv 2\pi \int_0^\pi d\theta \sin^3\theta F_1(\theta) \qquad (1)$$

For a Rayleigh scatterer, the Born approximation can be used, which leads to $F_1(\theta) \sim (V/k)\cos\theta$[42]. Multiple scattering of light in a disordered medium can be described by a diffusion tensor $D$, and Fick's diffusion law relates the photon current density $J$ defined per unit area to the local photon density gradient $\nabla\rho$:

$$J = -D \bullet \nabla\rho \qquad (2)$$

The magnetically induced off-diagonal component of the diffusion tensor $D$ generates the magnetic transverse current density $\triangle J_\perp$, which propagates perpendicular to the magnetic field **B** and the photon gradient $\nabla\rho$ and is mathematically denoted by $\Delta J_\perp \propto l_\perp \hat{B} \propto \nabla\rho$. The gradient $\nabla\rho$ is the result of the diffuse propagation of photons in the scattering medium and is not parallel to the incident beam. The parameter $l_\perp$ represents the coupling of photons within the scattering medium to the magnetic field. It is linearly proportional to the magnetic field and the Verdet constant $V$, and its dimension is the length. The difference between the upward and downward scattering magnetically induced photon currents is $\triangle I_\perp$, $\triangle I_\perp = I_U\text{-}I_D$, which is the integral of the magnetic transverse current density $\triangle J_\perp$ over the area of the photoelectric detector.

According to ref. 43, $D$ is expressed as

$$D_{ij}(\mathbf{B}) = D_0\delta_{ij} + D_H\varepsilon_{ijk}B_k + \Delta D_\perp(B^2\delta_{ij} - B_iB_j) + \Delta D_\parallel B_iB_j \qquad (3)$$

In the above equation, the first term represents the typical anisotropic scattering, the second term represents the magnetic transverse scattering that produces PHE, and the third and fourth terms represent the reluctance terms that produce PMR. For PMR, the relative transmittance decreases as the magnetic field increases, and we need to focus on whether the emission intensity of the RL decreases with the applied magnetic field. We additionally analyze in Supplementary Material III the emergence of the PMR in RLs in terms of absorption due to the imaginary part of the dielectric constant. Nevertheless, due to the presence of inherent intensity fluctuations, we need to exclude in the RL regime whether the intensity of the RL happens to decrease in a fluctuating trend when no magnetic field is added.

On the other hand, the presence of an external static magnetic field can eliminate the contribution of a single scattering event to the overall multiple scattering process[44]. As the magnetic field increases, the total scattering cross-section decreases[45]. During laser irradiation in a multiple-scattering media, the scattering intensity is caused by the interference between light from different scattering paths. Two components are essential: the individual scattering events, for example, characterized by the radiation pattern of each scatter, and the light transmission in the medium due to all the scattering events. In the presence of a magnetic field, both components are affected. For the propagation, this is the well-known Faraday effect (and Voigt or Cotton-Mouton effect), described by the correction of the complex refractive index by the magnetic field[1]. The imaginary part of the antisymmetric part of the dielectric constant can be interpreted as the Hall conductivity at optical frequencies when the multiple scattering in a disordered dielectric medium is exposed to an external magnetic field. In optical language, it mimics the "antisymmetric" extinction of the scattering mean free path. The real part of the antisymmetric term of the dielectric constant produces Faraday rotation and suppresses coherent scattering. The scattering cross-section measures the probability of occurrence of the scattering process, and a decrease in the

scattering cross-section implies a decrease in the number of scattering events. In the disordered scattering process, since each single scattering event is associated with the disordered features of the RL medium, the reduction of scattering events also decreases the overall scattering disorder of the RL system. The scattering mean free path $ls$ of RLs can be expressed as[46]

$$l_s(\omega) = \frac{1}{\rho\sigma_{sc}(\omega)} \qquad (4)$$

where $\rho$ is the density of scattering nanoparticles and $\sigma_{sc}$ is the scattering cross-section of a single nanoparticle with respect to frequency $\omega$. So the decrease of the scattering cross-section $\sigma_{sc}$ increases the scattering mean free path $l_s$ of RL.

An important aspect to emphasize here is that disordered multiple scattering is essential to photon diffusion in RLs. Indeed, when the scattering direction of a single scattering particle is laterally deflected, the overall RL scattering disorder is reduced. Thus, proving the scattering disorder reduction is an important aspect for probing the existence of PHE in an RL. It is actually very difficult to directly observe the scattering disorder of the RL, though it can be experimentally inferred from the emission and fluctuation of the RL spectrum. Here we analyze the degree of disorder in the random scattering from its connection with the RSB approach to the photonic glassy phase above the RL threshold.

The transition from the photonic paramagnetic to spin-glass phase with RSB arises as the result of the interplay between the randomness and nonlinear couplings of the RL modes. The statistical characterization of both regimes is performed through the analysis of the so-called Parisi's overlap parameter that essentially measures how intensity fluctuations of replicas of the RL system are correlated[31]. As previously noted, RL spectra exhibit random fluctuations across a range of wavelengths. Our focus is on assessing the random intensity fluctuations that exist between pulses in RLs. The spectral intensity fluctuations at a certain frequency, $\Delta_\alpha(k) = I_\alpha(k) - \bar{I}(k)$, are used to characterize the overlap between the intensity undulations of different replicas, where $I_\alpha(k)$ is the intensity of the light mode with replica index $\alpha$ and data collection index $k$, $\bar{I}(k)$ is the average over replicas of the intensity of each light mode,

$$\bar{I}(k) = \frac{1}{N}\sum_{\alpha=1}^{N_s} I_\alpha(k), \qquad (5)$$

$N$ is the number of replicas, and Parisi's replica overlap parameter[31] is defined as

$$q_{\alpha\beta} = \frac{\sum_{K=1}^N \Delta_\alpha(k)\Delta_\beta(k)}{\sqrt{\sum_{K=1}^N \Delta_\alpha^2(k)}\sqrt{\sum_{K=1}^N \Delta_\beta^2(k)}}. \qquad (6)$$

Based on the measured spectra, a total of $N(N\text{-}1)/2$ values of $q$ at each pumping energy are calculated to build the statistical distribution function $P(q)$ of $q_{\alpha\beta}$ values. Below the threshold, all RL modes oscillate independently during passive state operation, which corresponds to the paramagnetic state. Conversely, above the threshold, synchronous mode oscillation is impeded due to disorder, resulting in the photonic spin-glass phase. At low pump energies, the distribution $P(q)$ of the overlap parameter $q$ concentrates around zero with a Gaussian-like form, and the uncorrelated modes essentially do not interact in the paramagnetic phase. In contrast, above the RL threshold parameter $q$ distributes in the range $[-1, 1]$, with the emergence of two side maxima in $P(q)$ around $q = \pm1$, characterizing an RSB glassy regime in which the coherent oscillation of RL modes is hampered, and intensity fluctuations in distinct replicas are either correlated or anticorrelated. Here

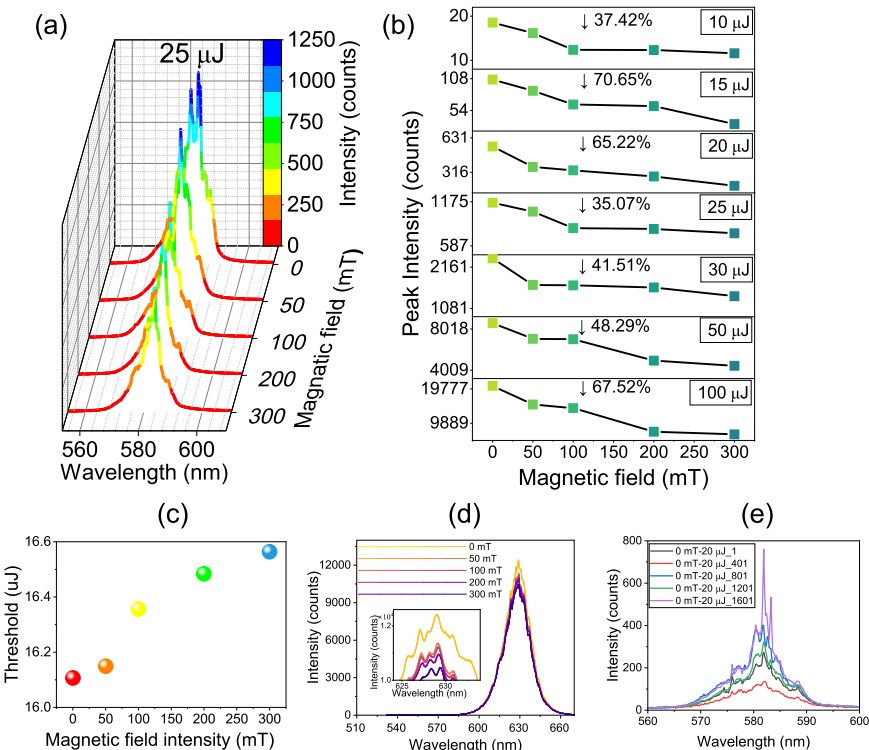

**Fig. 2 | Field-dependent RL spectroscopy symbolizing PMR in RL. a** RL spectra at the applied magnetic field intensity of 0, 50, 100, 200, and 300 mT at the pump energy of 25 μJ. **b** Variation of the maximum intensity of RL with the magnetic field at different pump energies. **c** Variation of the RL threshold with applied magnetic field intensity. **d** Fluorescence emission intensity at different applied magnetic field strengths under continuous laser irradiation of 632 nm Inset: partial magnification of the fluorescence spectrum. **e** RL spectra labeled as 1, 401, 801, 1201, and 1601, out of 2000 consecutive ones, acquired at the pump energy 20 μJ without applying a magnetic field.

we count the sum of the extreme values of the distribution $P(q)$ when $q_{ab}$ is positive and negative as

$$P(q)_{max} = P(q)_{max\,+} + P(q)_{max\,-} \qquad (7)$$

The RL threshold represents the critical point between the photonic paramagnetic and glassy phases, also signaling the appearance of the RSB phenomenon.

**Observation of PMR in RL and the field-dependent RL spectroscopy**

Spectra of RLs are presented in Fig. 2 and S4 for various magnetic field strengths and pump energies. Specifically, Fig. 2a displays the RL spectra for pump energies of 25 μJ while Fig. S4 displays the spectra for pump energies of 10, 15, 20, 30, 50, and 100 μJ. The measurements were taken at magnetic field strengths of 0, 50, 100, 200, and 300 mT for each pump energy. The spectrometer control software was used to collect RL spectra every 100 ms, with 2000 continuous measurements at different pump energy and magnetic field strength combinations. The RL spectra are generated by averaging the 2000 spectra. Results indicate that when no magnetic field is present (i.e., the magnetic field strength is 0 mT), the spectra transition from amplified spontaneous emission (ASE) at 10 μJ to spikes at 15 μJ and then to peaks at 20 μJ with increasing pump energy, demonstrating typical RL emission. As the pump energy continues to increase, the position of the RL peaks changes due to the disorder of RL scattering. This variation is caused by the disordered scattering of the RL. When the pump light enters the fiber, the laser dye provides gain, and the disordered multiple scattering of light between magnetic nanoparticles provides feedback. The nonlinear effects of saturation and mode competition will affect the

performance of the pump[47]. As the pump energy increases, the threshold of different modes will be reached, and new modes appear. General nonlinear theory is based on a self-consistent equation that determines how many modes exist in a fixed pump energy and the frequencies of these modes.

We observe in Fig. 2 that the intensity of both ASE and RL regimes decreases as the magnetic field strength increases in the seven sets of pump energies tested. Figure 2b displays the decreasing trend in peak intensity of the RL at each pump energy with increasing magnetic field strength, which shows the PMR effect in RLs. Near threshold, the RL intensity decreases by ~70% when the applied magnetic field is 300 mT. At sub-threshold and suprathreshold, the RL intensity decreases by >30%, and when the pump energy is much higher than the threshold, the energy decreases by a degree of more than 60% again. This suggests that the phenomenon of PMR in the POF RL system is more pronounced near and far above the threshold. The intensities of RL at different pump energy and magnetic field intensities are depicted in Fig. S4g, with the inflection point of the fitted line corresponding to the threshold. The inset in Fig. S4g is the magnification of the inflection point. The increase of applied magnetic field results in an increased RL threshold as shown in Fig. 2c. The first explanation for the PMR effect in RLs is based on the magnetic dielectric effect, where the electrons in each $Fe_3O_4@SiO_2$ NP within the MGPOF become polarized in the presence of a constant magnetic field, leading to an increase in their dielectric constants[48,49]. We elaborate on this interpretation in Supplementary Information III.

In this experiment, we utilized a 635 nm continuous laser (Changchun New Industries Optoelectronics, MGL-V-532-2W) to investigate whether the output transmitted light intensity changed with the magnetic field when this laser was injected into the same fiber

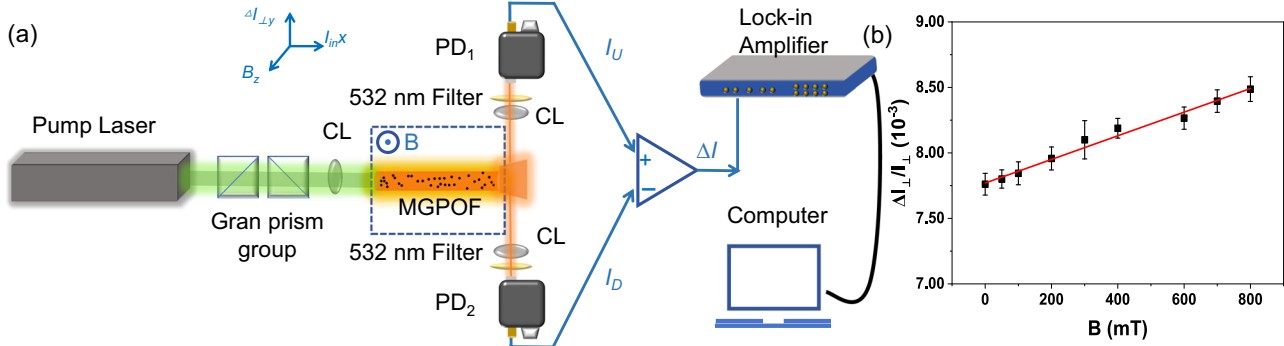

**Fig. 3 | The magnetic transverse photocurrent symbolizing PHE in RL. a** Schematic setup for the observation of the PHE in RL. CL convergent lens, PD photoelectric detector. **b** Normalized magneto-transverse light current $\triangle I_\perp/I_\perp$ versus magnetic field. The error bars show the standard error of $\triangle I_\perp/I_\perp$ fluctuations.

as described above. RL can only be produced under pulsed laser excitation at 532 nm but not under continuous laser excitation at 635 nm. As depicted in Fig. 2d, an increase in the magnetic field resulted in decreased transmitted light intensity, proving that an increase in the magnetic field enhances the absorption and makes the intensity of the emission light decrease. It is worth noting that due to the inherent randomness of the RL, the intensity of its spectrum and the position of the peak fluctuate randomly in the limit of the gain medium and scattering intensity. To demonstrate the magnetic field's modulating effect on RL intensity, Fig. 2e displays the 1st, 401st, 801st, 1201st, and 1601st of 2000 sets of continuously collected RL spectra without magnetic field at 20 μJ pump energy. The RL intensity gradually decreases when the magnetic field increases, while the RL intensity fluctuates irregularly when all other experimental conditions are the same and only the magnetic field is removed. The erratic RL intensity fluctuations rule out the attenuation of RL intensity by time-induced bleaching of the laser dye, which supports the modulating effect of the magnetic field on RL intensity.

**Observation of PHE in RL**

Experiment on direct observation of the PHE in RL is shown in Fig. 3a. The pump laser is coupled into one end of the MGPOF, the generated RL and the residual pump laser exit from the other end. The magnetic field generated by the magnetic field generator (z axis direction) is perpendicular to the MGPOF (x-axis direction), and the MGPOF is completely within the magnetic pole range of the magnetic field generator. RL is collected by two PDs outside the magnetic field area and at the upper and lower positions (y axis direction) of the exit end face of the fiber. Filters are placed in front of PD1 and PD2 to filter the pump light. PD1 and PD2 are connected to the lock-in amplifier after the differential circuit to detect the magnetic transverse photocurrent $\triangle I_\perp = I_U - I_D$. $I_U$ and $I_D$ are the light current tested by PD1 and PD2, respectively. It is then normalized by the transverse scattering intensity $I_\perp = I_U + I_D$. The ratio $\triangle I_\perp/I_\perp$ is used to describe the magnitude of the magnetic transverse photocurrent, which plays the same role as the Hall angle in the magnetic transport of electrons. The results shown in Fig. 3b confirm the linear dependence of the magnetic transverse photocurrent generated by the PHE on the magnetic field strength in a RL system. It is worth noting that the $Fe_3O_4@SiO_2$ magnetic nanoparticles are randomly distributed in the core of the MGPOF, so the MGPOF does not have perfect spatial symmetry. Therefore, a "net zero" lateral deflection of RL emission at any given point along the MGPOF should not be expected. When there is no applied magnetic field, $\triangle I_\perp/I_\perp \neq 0$. As the intensity of the applied magnetic field B increases, the magnetic transverse photocurrent increases. This linear dependence of the magnetic transverse photocurrent on the applied magnetic field shows the occurrence of PHE in the RL. From Eq. (1) we conclude that the degree of PHE in the RL based on MGPOF is the slope

of the red fitting curve $\eta = 1.099 \times 10^{-9}/mT$, which is the evidence of PHE in RL systems.

**Field-dependent RSB in RL**

Figure 4 and Fig. S5 illustrate the distributions $P(q)$ of the Parisi overlap parameter for the correlations of RL intensity fluctuations at various magnetic field strengths and pump energies. Specifically, Fig. 4 displays the distribution obtained from the RL spectra for pump energies of 10, 15 25, and 50 μJ at 0, 50, 100, 200 and 300 mT. The three columns in Fig. S5 are for pump energies of 20, 30, and 100 μJ, respectively. The overall distribution $P(q)$ exhibits narrow Gaussian-like distributions when the pump energy is 10 μJ below the threshold. However, as the pump energy increases to just above the threshold value of 15 μJ, $P(q)$ changes significantly towards the double-peaked profile that characterizes the photonic RSB glassy phase in the RL regime. At 25 μJ pumping, the extreme values of $P(q)$ are smaller, and a Gaussian-like form takes place at very high pumping energies of 50 μJ and 100 μJ. For very high pumping energies, the degree of disorder within the POF is greatly diminished and the replica-symmetric regime is recovered. Hence the profile of $P(q)$ returns to the Gaussian-like form[36,50–52].

The impact of the magnetic field on the statistical distribution of correlations between RL intensity fluctuations can also be investigated. Figure 5 illustrates the distributions $P(q)$ under different magnetic field strengths, ranging from 0 to 300 mT at different pump energies. A Gaussian function is fit to each distribution to determine the standard deviation $\sigma$ for 10, 30, 50, 100 μJ. Interestingly, with increasing magnetic field strength, the standard deviation of the Gaussian distribution is reduced by more than 80% at 30 and 50 μJ. The decrease in the standard deviation with increasing magnetic field strength indicates that the scattering disorder of RL intensity fluctuations is attenuated by the orienting effect of magnetic particles on the magnetic field. This indicates a significant magneto-optical regulation in this pumping regime of the RL system based on POFs. The third column of Fig. S5 show a similar situation of the standard deviation $\sigma$ at pump energies of 100 μJ, which reduces by 55%. At the pumping energy of 15 μJ, $P(q)_{max}$ gradually increases by about 75% as the magnetic field increases, while the dispersion of the overall distribution $P(q)$ reduces with a higher magnetic field when the pump energy is marginally above the threshold. The increase in the magnetic field results in a reduction in the disorder effect on the correlation of RL intensity fluctuations thanks to the orientation of the magnetic particles brought about by the magnetic field, which weakens scattering disorder. A similar increase in $P(q)_{max}$ is observed when the pump energy is 20 μJ and 25 μJ, which increases by around 162% and 21%. It is noteworthy that at 10 μJ, the distribution $P(q)$ shifts toward $q = 0$ when compared to 15 μJ and 20 μJ. When the pumping energy is further increased to 30 μJ, $P(q)$ returns to an approximate

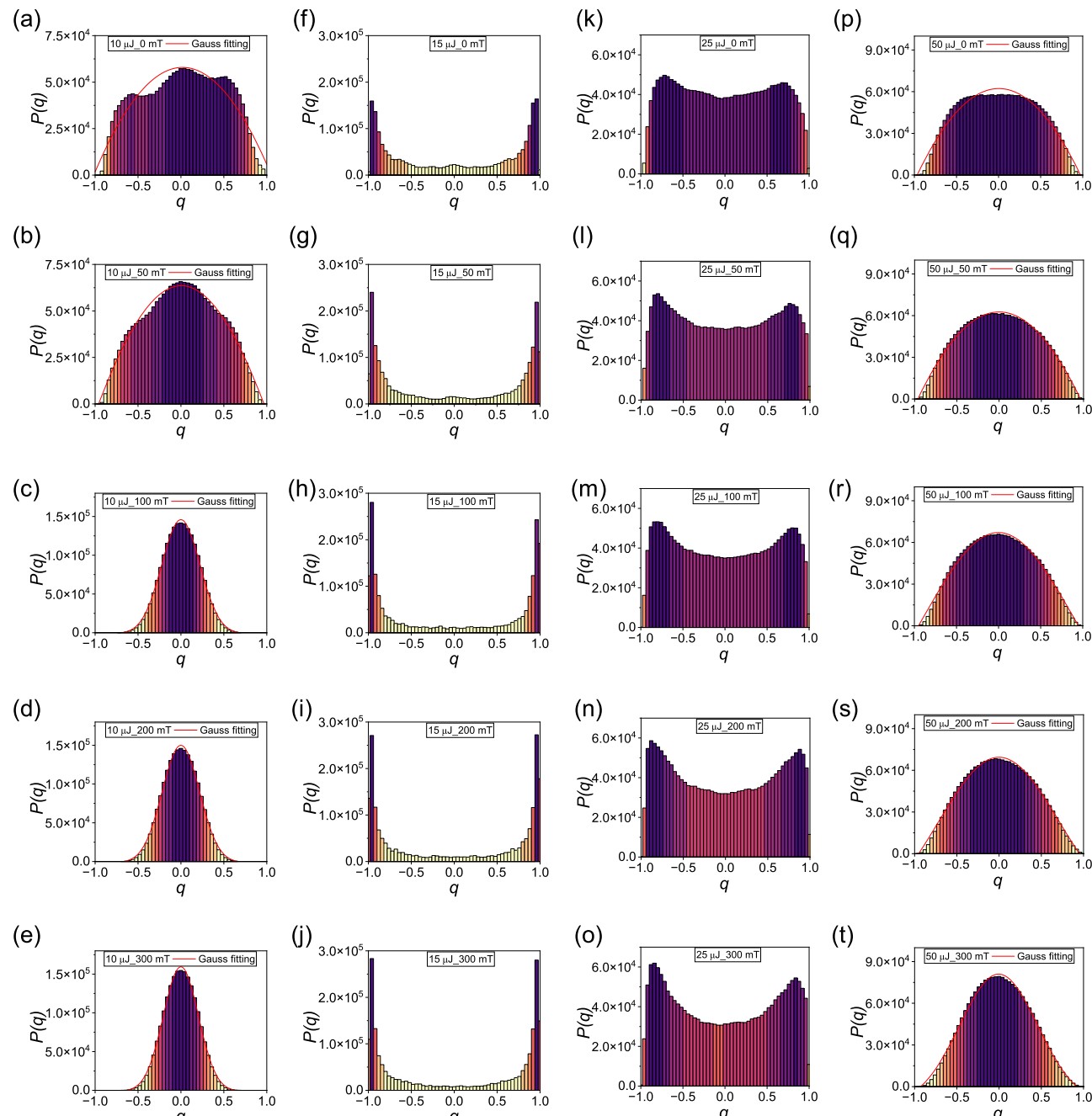

**Fig. 4 | Overlap distributions signalizing the photonic replica-symmetric para-magnetic to RSB glassy transition.** Plots of the distribution $P(q)$ of the Parisi overlap parameter $q$ for different pumping energies. 10 µJ: (below the threshold, **a**–**e**); 15 µJ: (around the threshold, **f**–**j**); 25 µJ: (above the threshold, **k**–**o**) and 50 µJ: (**p**–**t**); at applied magnetic field strength of 0 mT (the first row), 50 mT (the second row), 100 mT (the third row) 200 mT (the fourth row) and 300 mT (the fifth row).

Gaussian distribution as in the second column of Fig. S5. All these phenomena indicate that the increase in the magnetic field enhances the orientation effect on the magnetic particles, which then weakens the scattering disorder.

We should remark that the energy threshold also increases with the field, as shown in Fig. 2c. So, the increase of $P(q)^{max}$ with the magnetic field in Fig. 5 cannot be solely used to directly infer the efficiency of the RL, since the effect of the proximity of the threshold must be also taken into account in the analysis. To be more specific, consider, for example, the case with excitation energy 15 µJ in Fig. 4f–j. Due to the dependence of the threshold on the field, we notice in Fig. 4g for 300 mT that the input energy 15 µJ is much closer to the threshold than

in Fig. 4f, in which no field is applied. It is thus justified that $P(q)^{max}$ is larger in Fig. 4g than in Fig. 4f (with $P(q)$ more concentrated around the extrema $q = \pm 1$ in Fig. 4g), as actually observed. The same reasoning also applies to higher magnetic fields in Fig. 4h–j with 15 µJ, and for other values of excitation energy as well.

These RSB findings are also consistent with the analysis of the correlation coefficient $r$ $(\lambda 1, \lambda 2)$ between intensity fluctuations of modes at distinct wavelengths in the same replicas at each pump energy and magnetic field intensity[53–55] in Supplementary Information V. We also probe the findings and analysis of RSB transition and correlation coefficient with numerical simulation results based on scattering theory in Supplementary Information VI.

(a)

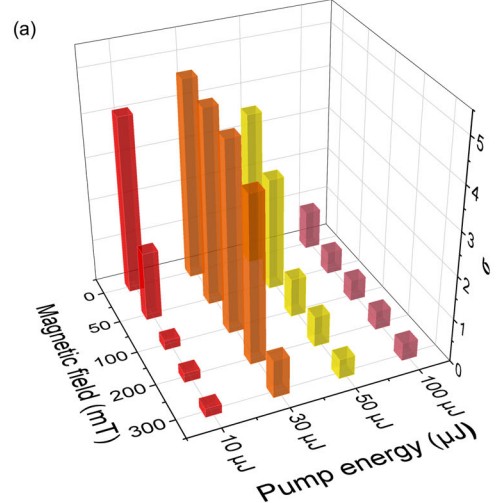

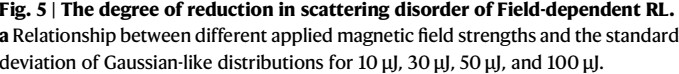

(b)

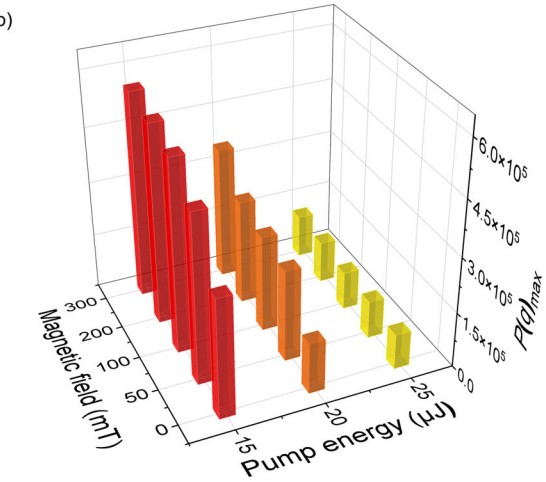

**Fig. 5 | The degree of reduction in scattering disorder of Field-dependent RL. a** Relationship between different applied magnetic field strengths and the standard deviation of Gaussian-like distributions for 10 μJ, 30 μJ, 50 μJ, and 100 μJ.

**b** Relationship between different applied magnetic field strengths and $P(q)_{max}$ of the five $P(q)$ distributions of 15 μJ, 20 μJ, and 25 μJ.

## Discussion

This work reveals the presence of PHE and PMR in a field-dependent RL. A typical coherent RL is generated based on a MGPOF doped with magnetic $Fe_3O_4$@$SiO_2$ NPs. Macroscopically, it is directly observed that the applied magnetic field causes the RL to generate magnetic transverse photocurrent, and the increase of the magnetic field intensity leads to the decrease of RL intensity. We show the relationship between the magnetic transverse photocurrent and the magnetic field strengths. We introduce a fresh perspective on magnetic field detection and sensing applications. By utilizing the PHE or PMR, RLs with micro-nano structures can serve as effective tools for magnetic field sensing. These compact detectors are not only cost-effective but also easy to integrate into existing systems.

In addition, to explore the influence of these two effects on the RL microscopic scattering process, we analyze the effect of the applied magnetic field on the radius of the ordered spins of the magnetic dipole and the scattering cross-section of RL. From the variation of RSB statistical distribution in the RLs spin-glass theory, it is found that the distribution $P(q)$ of the Parisi overlap parameter exhibits reduced dispersion with increasing magnetic field, indicating a decrease in the scattering disorder of the RL. Numerical simulations based on scattering theory further validate these analyses. Overall, this work connects microscopic magnetic particle scattering with the macroscopic magneto-optical effect on the modulation of RL, and proposes a means of regulating disorder in RL multiple scattering. Our findings offer distinct insights into the modulation of RL scattering feedback.

## Methods
### Experimental setup
Based on the above analysis, we design experiments to analyze the effect of the applied magnetic field on the RL emission intensity and its fluctuations. The experimental setup for pumping an MGPOF with a pulsed laser (Quantel, Q-SMART450) at a wavelength of 532 nm is illustrated in Fig. 3. The pumped light is injected into the POF through a Gran laser prism, which controls the pump energy. The pump laser is coupled into one end of the MGPOF, and the generated RL and the residual pump laser exit from the other end so that the RL is given by the entire MGPOF. To prevent residual pump light from affecting subsequent test results, the exiting light is filtered, and only the excitation light is received by the spectrometer probe after convergence. To investigate the impact of the magnetic field on RL intensity

fluctuations, the MGPOF is exposed to a magnetic field generator (CH-Hall Electronic Devices Inc.) whose magnitude is regulated via a DC power supply. Additionally, a Gauss meter positioned above the POF measures the magnetic field strength.

## Data availability
The data that support the findings of this study are available from the corresponding author upon request. Source data are provided in this paper. Source data are provided with this paper.

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

## Acknowledgements

We acknowledge funding support from National Natural Science Foundation of China (grant nos. 12174002; 11874012, 11874126, and 51771186), Excellent Scientific Research and Innovation Team of Anhui Province (grant no. 2022AH010003), Key Research and Development Plan of Anhui Province (grant no. 202104a05020059), Innovation project for the Returned Overseas Scholars of Anhui Province (grant no. 2021LCX011), The University Synergy Innovation Program of Anhui Province (grant no. GXXT-2020-052); Anhui Project (grant no. Z010118167). ASLG and EPR thanks the funding from CNPq and FACEPE (Brazilian agencies). The authors appreciate Prof. Dr. Yonghua Lu (University of Science and Technology of China) for the fruitful discussion.

## Author contributions

W.D. and Z.H. initiated the idea. W.D. and L.H. conducted the numerical simulations. W.D. and J.X. fabricated the samples. W.D., Z.H., F.X., and Y.L. performed the measurements. W.D., Z.H., S.L., Y.K., Z.C., K.Z., E.P.R., and A.S.L.G. prepared the manuscript. Z.H., E.P.R., A.S.L.G., L.Z., K.X., and B.Y. supervised the project. All the authors discussed and analyzed the results.

## Competing interests

The authors declare no competing interests.
