## [Peer Review File · Nature Communications]

REVIEWER COMMENTS

Reviewer #1 (Remarks to the Author):

The photonic Hall effect manifests itself as the deflection of light perpendicular to the incident beam and transverse magnetic field. This phenomenon can be seen as the photonic version of Hall effect in electronic systems and its physical origin is a magnetically induced change in the optical scattering properties of a single particle in a disordered medium. In the present work, the authors have introduced the field-dependent theory of magnetic disorder in scattering nanoparticles for predicting the photonic Hall effect in random lasers and demonstrated it experimentally with the Fe₃₀O₄@SiO₂ material system. Moreover, they have also proposed and verified the corresponding photonic magnetoresistance. This is the first experiment observation of photonic Hall effect and photonic magnetoresistance in random lasers. Generally, the results discussed here are interesting and reliable. However, some major concerns should be addressed carefully before the possible publication in Nature Communications.

In addition to the photonic Hall effect, the photonic magnetoresistance is also a very important part of the present work. Thus, the title "Observation of the photonic Hall effect in random lasers" should be modulated accordingly. Has the photonic magnetoresistance been observed in random lasers before?

Can the authors provide a clear and simple physical picture for describing the relationship between the photonic Hall effect, photonic magnetoresistance, and random lasers? For example, as for Fig. 1, we cannot directly get the main idea of the present work unless we read the legend and main text carefully. For this point, it does not meet the high standard of Nature Communications.

It seems to me that the observation of photonic Hall effect in this work is dependent on an indirect measurement. Is it possible to observe the photonic Hall effect (just like trajectory deflection of light beam in this system) directly? It is very similar to the observation photonic spin Hall effect in metasurfaces. For example, we can directly observe the spin-dependent splitting of left- and right-handed circularly polarized components when a linearly light beam is reflected or transmitted from the medium. See the following papers:

"Photonic Spin Hall Effect at Metasurfaces" Science 339, 1405 (2013)

"Spin-optical metamaterial route to spin-controlled photonics" Science 340, 724 (2013)

The authors should add more contents for discussing the potential or practical applications by exploring the photonic Hall effect and photonic magnetoresistance in random lasers.

Reviewer #2 (Remarks to the Author):

The manuscript "Observation of the photonic Hall effect in random lasers" by Wenyu Du et al. reports on the modulation of random laser emission by applying a magnetic field to the proposed fabricated disordered fiber system.

I have the following concerns:

1. I cannot understand the sample composition and functionality. Ferromagnetic disordered NPs have the structure reported in Fig. 1. They have a core of Fe₃O₄ and a shell of SiO₂. The whole particle scatters light and the scattering strength is dependent on the applied magnetic field as the theoretical calculations show. What is the meaning of Figure 1? What are the arrows for? I understand that the white ones represent the magnetic moment of Fe₃O₄ part, but I don't understand what is spin disorder. Is it a physical quantity?

Moreover, it is not clear to me what is the final structure of the fibers. Are they made of NPs packed inside a polymeric matrix? Is it a solid structure? Is the random laser given by the entire fiber? What is the length along the excitation beam? Has the length of the fiber any role?

I think more clarification about sample preparation and realization is needed and I suggest putting everything in one paragraph and not dispersed inside the main text, the methods and the supplementary sections. I find Figure 1 misleading and not relevant, maybe there is also some mistake in the caption (up and down instead of right and left).

2. The main result of the manuscript is the tuning of the emission threshold by applying a magnetic field to the NPs. The light scattering is decreased by increasing field and the random laser gets less efficient and this is for sure a nice way for the modulation of RLs.

The authors demonstrate this by spectral (figure 2) and replica symmetry breaking (figure 3) analyses. The latter one has been proposed in many other works on different systems as a powerful tool to show RL action in glassy RLs with fluctuating emission.

Can the authors explain why in Figure 3 d-I, the RL results more efficient by increasing the magnetic field and thus by decreasing the scattering strength? Isn't it the opposite of their major claim?

A deep discussion is required on this point.

3. To my opinion, spectral correlations (equation 5 and Figure 4) do not add any useful information. They confirm the results obtained by the spectral and RSB analysis.

Concluding, I think the many data reported in the manuscript, including supplementary material, are redundant and can be summarized in few significative ones. The only novel point of the manuscript is the ability to tune the RL emission and so the material structure and fabrication. The data analysis is not novel, the equations are well known and the theoretical part is not new, although useful.

I think after substantial revision in the material description, data presentation and organization, the manuscript can be published in a journal more specialized in optics/photonics.

Reviewer #3 (Remarks to the Author):

This manuscript presents an experimental study of a magnetic random laser under the influence of a

strong applied magnetic field. Scatterers have diameters of tens of nanometers and are composed of magnetite with a shell of silicon dioxide [S1]. Optical gain is provided by laser dye doped inside an optical fiber core along with the nanoparticle scatterers, which are presumably randomly placed. Core diameter is 32 μm and cladding diameter 674 μm . Experiments are supported by scattering theory presented in the supplementary information, which shows the scattering coefficient decreasing with an increasing magnetic field as well as absorption and extinction increasing with an increasing magnetic field. Both effects should increase the lasing threshold, which is observed explicitly in Fig. 2(d). An increase of the threshold also means that the applied magnetic field is inversely related to the pump energy. Indeed, as an example in Fig. 4, increasing the magnetic field from 0 to 300 mT in (d) to (f) alters the heatmap to resemble that seen at lower pump energy in (a). However, as discussed in Refs. [1-3], the photonic Hall effect (PHE) and photonic magnetoresistance (PMR) depend specifically on the generation of a transverse light current. While it may be possible that such a current is generated in these experiments, there is not even a discussion of this phenomenon (for experimental observation see [2]). Moreover, replica symmetry breaking itself is not dependent on a transverse current. Therefore, while this is an interesting study on the effect of magnetism in random lasers, there is no conclusive evidence presented of either the PHE or PMR. We believe the PHE and PMR could be speculatively discussed in the Conclusion of a revised manuscript, if the authors wish, but stating that this is the first observation of these phenomena in random lasers is misleading. There are also several questions that should be addressed, which are discussed below.

Equation (1),

The contribution of light scattering to the experimental results is supported by the supplementary information even though it appears much smaller than contributions from absorption and extinction. However, just because light is being scattered differently with a magnetic field applied and altering the mean free path does not mean a transverse current has been established, which is a requirement for the PHE and PMR. Also, note that the quasimodes of random multiple-scattering systems are spatially random and light in the random fiber is already being deflected laterally (see Fig. 1 of [18]).

Figure 1,

The diagrams here are confusing. From [S1], we believe the composition of nanoparticles is a silicon dioxide shell around a magnetite core. But in this figure, silicon dioxide is shown only on the bottom half of the scatterers. More importantly, below roughly 70 nm, magnetite nanoparticles have a single magnetic domain [Nature Sci Rep 7, 9894 (2017)]. This contradicts the authors' description of an external magnetic field aligning a distribution of magnetic moments in a single nanoparticle. Also, the placement of these scatterers in the fiber core, which is presumably random, is not discussed. There are a few types of disorder here that should be presented more clearly: (i) the amorphous nature of the silicon dioxide shell on each nanoparticle, (ii) the initial random orientation of the single-domain magnetic moment of each nanoparticle before a magnetic field is applied, and (iii) the spatially random placement of the nanoparticles in the optical fiber core. This figure requires significant revision.

Lines 249-250, "The RL spectra are generated by averaging the 0th, 40th, 801st, 1201st, and 1601st spectra."

Why are these particular instances chosen? Just five spectra is statistically irrelevant and raises the question as to how data in Figs. 2(c), 2(d), and 3 are calculated. Hopefully all 2000 measurements were

used in those calculations -- please clarify.

Lines 258-259, "This feature can form a scattering closed loop when light is scattered between particles because the scattering disorder is strong enough."

Random laser modes formed by underlying quasimodes due to multiple scattering are more complex than can be described by just a "scattering closed loop" [e.g., Adv Opt Photon 3, 88 (2011)].

Figure 2(c),

Peak Intensity is shown in units of counts but the scale has a maximum of 1, which differs from what is shown in Figs. 2(a-c,f).

Line 307,

What is meant by "time-induced weakening"? Bleaching of the laser dye?

Lines 322-324, "For very high pumping energies, the degree of disorder within the POF is greatly diminished and the RL modes oscillate coherently under nonlinear action."

This statement appears to conflict with Figs. 4(j)-4(l), where significant anti-correlations exist along with the absence of correlations between RL modes.

Figure 3(f,i),

The values of $P(q)_{\max}$ do not seem to match the values observed in the $P(q)$ distributions, e.g., Fig. 3(d) shows $P(q) < 2e5$ but the value in Fig. 3(f) at $H = 0$ is above $3e5$. Such discrepancies are seen in the supplementary information as well.

Figure 4(a-c),

What is the physical origin of the periodic, off-diagonal stripes? They are not random lasing modes, since the pump energy is below threshold. They cannot stem from the random quasimodes either, which are at random wavelengths (not regularly spaced).

Various errors in referencing other sections exist throughout the manuscript, e.g., should lines 290 and 436 refer to section VI of the supplementary information?

Finally, we suggest including additional results at a higher applied magnetic field to match the hysteresis curve in the supplementary information. The optical response should qualitatively follow the hysteresis curve, i.e., at a high enough applied magnetic fields there should be no noticeable changes, meaning all magnetic effects have saturated.

Response to the reviewers' comments

Reviewer #1 (Remarks to the Author):

The photonic Hall effect manifests itself as the deflection of light perpendicular to the incident beam and transverse magnetic field. This phenomenon can be seen as the photonic version of Hall effect in electronic systems and its physical origin is a magnetically induced change in the optical scattering properties of a single particle in a disordered medium. In the present work, the authors have introduced the field-dependent theory of magnetic disorder in scattering nanoparticles for predicting the photonic Hall effect in random lasers and demonstrated it experimentally with the $\text{Fe}_3\text{O}_4/\text{SiO}_2$ material system. Moreover, they have also proposed and verified the corresponding photonic magnetoresistance. This is the first experiment observation of photonic Hall effect and photonic magnetoresistance in random lasers. Generally, the results discussed here are interesting and reliable. However, some major concerns should be addressed carefully before the possible publication in Nature Communications.

Our response: We are grateful to Reviewer #1 for investing time and expertise in reviewing our manuscript. We also thank the reviewer for the clear description of our manuscript. We are glad about the positive evaluation.

1. In addition to the photonic Hall effect, the photonic magnetoresistance is also a very important part of the present work. Thus, the title “Observation of the photonic Hall effect in random lasers” should be modulated accordingly. Has the photonic magnetoresistance been observed in random lasers before?

Our response: We thank the reviewer for these important suggestions, we agree that photonic magnetoresistance is also an important part of this article. To the best of our knowledge, PMR has not been observed in RLs before. Our paper is the first to explore

this phenomenon in RLs. So, following the reviewer's suggestions, we have modified the title of the manuscript accordingly. The new title is: "Observation of the photonic Hall effect and photonic magnetoresistance in random lasers".

2. Can the authors provide a clear and simple physical picture for describing the relationship between the photonic Hall effect, photonic magnetoresistance, and random lasers? For example, as for Fig. 1, we cannot directly get the main idea of the present work unless we read the legend and main text carefully. For this point, it does not meet the high standard of Nature Communications.

Our response: We thank the referee for the careful insight. We have considerably modified Fig. 1 to better represent the observed effects. As for the physical picture underlying these phenomena, we kindly refer the reviewer to the beginning of this response, in which the relationship between the photonic Hall effect, photonic magnetoresistance, and RLs has been discussed.

3. It seems to me that the observation of photonic Hall effect in this work is dependent on an indirect measurement. Is it possible to observe the photonic Hall effect (just like trajectory deflection of light beam in this system) directly? It is very similar to the observation photonic spin Hall effect in metasurfaces. For example, we can directly observe the spin-dependent splitting of left- and right-handed circularly polarized components when a linearly light beam is reflected or transmitted from the medium. See the following papers:

"Photonic Spin Hall Effect at Metasurfaces" Science 339, 1405 (2013)

"Spin-optical metamaterial route to spin-controlled photonics" Science 340, 724 (2013)

Our response: It is a very good question raised by the referee. It is indeed possible to observe the photonic Hall effect directly, as we actually show in the new Fig. 3 of the revised manuscript (see, also, the beginning of this response above). We thank the reviewer for this nice suggestion and for listing these seminal articles. The photonic Hall effect and the photonic spin Hall effect are not the same. The specific distinctions

and our supplementary direct observation experiments are at the beginning of this response.

4. The authors should add more contents for discussing the potential or practical applications by exploring the photonic Hall effect and photonic magnetoresistance in random lasers.

Our response: We thank the reviewer for these important suggestions. The methods mentioned in the manuscript provide a new perspective for magnetic field detection and sensing applications. The gain scattering system of RLs can be made into very small structures to detect magnetic fields (Wiersma D, “The smallest random laser”, *Nature*, 406, 133-135 (2000)). Based on the photonic Hall effect or the photonic reluctance principle, the RL with micro-nano structure can be used for magnetic field sensing. Such magnetic field detectors are small in size, extremely low in cost, and easy to integrate. We show the relationship between the magnetic transverse photocurrent and the magnetic field strengths. The optical signals emitted by RLs are converted into electrical signals using two PDs at the upper and lower positions of the MGPOF exit end. Based on PHE, the magnitude of the applied magnetic field at the location of the MGPOF can be tested.

In this paper, the regulation process of magneto-optic effect on RL is comprehensively demonstrated from the micro and macro perspectives. From the macroscopic phenomenon, the magnetic transverse photocurrent caused by the applied magnetic field corresponds to the photon Hall effect in the RL system. The emission intensity of the RL is weakened by the external magnetic field, which corresponds to the photon reluctance phenomenon in the RL system. From the microscopic phenomenon, the external magnetic field makes the ferrite magnetic scattering nanoparticles overcome the structural disorder on the surface, and the radius of the ordered magnetic moment increases, which weakens the RL scattering disorder and then weakens the RL intensity fluctuation disorder. Therefore, this manuscript comprehensively shows the regulation effect of photon Hall effect and photon reluctance on RL emission performance, and proposes a method to regulate RL scattering disorder. The microscopic regulation

process of RL scattering by magnetic field is visualized to the macroscopic RL intensity fluctuation disorder.

Modifications in the revised manuscript: [page 19 in Conclusion]

We show the relationship between the magnetic transverse photocurrent and the magnetic field strengths. We introduce a fresh perspective on magnetic field detection and sensing applications. By utilizing the PHE or PMR, RLs with micro-nano structures can serve as effective tools for magnetic field sensing. These compact detectors are not only cost-effective but also easy to integrate into existing systems.

In addition, to explore the influence of these two effects on the RL microscopic scattering process, we analyze the effect of the applied magnetic field on the radius of the ordered spins of the magnetic dipole and the scattering cross-section of RL. From the variation of RSB statistical distribution in the RLs spin glass theory, it is found that the distribution $P(q)$ of the Parisi overlap parameter exhibits reduced dispersion with increasing magnetic field, indicating a decrease in the scattering disorder of the RL. Numerical simulations based on scattering theory further validates these analyses. Overall, this work connects microscopic magnetic particle scattering with the macroscopic magneto-optical effect on the modulation of RL, and proposes a means of regulating disorder in RL multiple scattering. Our findings offer new insights into the modulation of RL scattering feedback.

Finally, we thank once again the reviewer for the quite nice suggestions and keen comments. We have made an effort to address every point raised and make all modifications suggested in the revised manuscript, in a sufficiently clear way for the benefit of a broad readership. We therefore hope that, with these significant improvements, the referee can now consider our manuscript suitable for publication in *Nature Communications*.

Reviewer #2 (Remarks to the Author):

The manuscript “Observation of the photonic Hall effect in random lasers” by Wenyu Du et al. reports on the modulation of random laser emission by applying a magnetic field to the proposed fabricated disordered fiber system.

I have the following concerns:

Our response: We are grateful to Reviewer #2 for investing time and expertise in reviewing our manuscript.

1. I cannot understand the sample composition and functionality. Ferromagnetic disordered NPs have the structure reported in Fig. 1. They have a core of Fe₃O₄ and a shell of SiO₂. The whole particle scatters light and the scattering strength is dependent on the applied magnetic field as the theoretical calculations show. What is the meaning of Figure 1? What are the arrows for? I understand that the white ones represent the magnetic moment of Fe₃O₄ part, but I don't understand what is spin disorder. Is it a physical quantity?

Our response: We start by remarking that we now present a completely new Figure 1, with a much clearer explanation, in the revised manuscript (see, also, the beginning of this response).

The new Fig. 1 illustrates the structure and field-dependent magnetic NP morphology. The spontaneous, noncorrelated spin disorder at the particle surface is strongly related to structural surface disorder. With an increasing magnetic field, the collinear magnetic volume overcomes the structurally coherent particle size. Arrows indicate magnetic spins, white arrows indicate collinear magnetic dipole spins in the core, and purple arrows indicate disordered magnetic spins on the nanoparticle surface. The surface spin disorder of magnetic nanoparticles is an inherent property of nanomaterials. The existence of spin disorder on the particle surface is confirmed in the work [Disch S,

Wetterskog E, Hermann R P, Wiedenmann A, Vainio U, Salazar-Alvarez G Bergström L and Brückel T, “Quantitative Spatial Magnetization Distribution in Iron Oxide Nanocubes and Nanospheres by Polarized Small-Angle Neutron Scattering”, *New J. Phys.* 14, 013025 (2012)].

Modifications in the revised manuscript: [the second paragraph on page 7]

From the microscopic disordered multiple scattering during RL generation, we note that the presence of an applied magnetic field causes a significant increase in the magnetic moment of ferrite nanoparticles³⁸. The applied magnetic field can overcome the disorder of the ferrite nanoparticle structure surface, gradually polarizing the uncorrelated, disordered surface spins, as shown in the pink box of Fig. 1. The upper semicircle represents the structural morphology and the lower semicircle represents the magnetic morphology. Prior to the application of a magnetic field, the structural and magnetic nanoparticles are equal in size as shown in the circle on the left, whereas the initially disordered surface spin is gradually polarized in the applied magnetic field, causing the magnetic radius to increase beyond the structurally disordered surface region as shown in the circle on the right.

The deep pink squares show the structurally coherent grain of Fe_3O_4 core, the light pink dots show the structural disorder of SiO_2 shell region. The silica shell on each nanoparticle has its amorphous nature. The white arrows represent the collinear magnetic dipole spin in the magnetic core and the purple arrows represent the spin disorder of amorphous SiO_2 shell. The overall magnetic moment of scattering nanoparticles, i.e., the core-shell structure $\text{Fe}_3\text{O}_4@\text{SiO}_2$ magnetic NPs, increases with the magnetic field. Following the Faraday effect, the magnetic field causes the rotation of the scattering polarization plane, which is linearly proportional to the component of the magnetic field toward the direction of light wave propagation. An increase in the applied magnetic field leads to an enhanced effect on the orientation of scattered light, causing an increase in the radius of the ordered magnetic dipole moment. This results in a weakened scattering disorder of RL, which ultimately reduces the RL intensity fluctuations.

Moreover, it is not clear to me what is the final structure of the fibers. Are they made of NPs packed inside a polymeric matrix? Is it a solid structure? Is the random laser given by the entire fiber? What is the length along the excitation beam? Has the length of the fiber any role?

Our response: The final structure of the fiber is a cylindrical polymer optical fiber waveguide. Figs. S1(d) and S3 show the microscopic and real images, respectively. Also, please notice below in green the modified text in the revised manuscript, with further experimental details, in the line suggested by the reviewer.

Specific answers to the questions above are as follows: Yes, we doped laser dye and the magnetic scattering NPs in the fiber core. We describe the fabrication process of the magnetic gain POF in Supplementary Information I.

Yes, it is a solid structure.

Yes, the pump laser is coupled into one end of the fiber, the generated RL and the residual pump laser exit from the other end. The experimental setup is shown in Fig. S3.

The length of the POF is 5 cm.

The length 5 cm is consistent with the diameter of the magnetic pole, so the whole fiber RL is placed in the magnetic field environment.

Modifications in the revised manuscript: [the first paragraph on page 3 in Supplementary information]

First, a mold is assembled for the fiber optic prefabricated rod with a transparent Teflon tube with a diameter of 2 cm inserted into both ends of a homemade base, and a Teflon rope with a diameter of 0.8 mm is fixed at the center of the Teflon tube. Dilauroyl peroxide (LPO) with a mass fraction of 0.22% was added to the beaker as the initiator, followed by methyl methacrylate (MMA) and butyl acrylate (BA) in a volume ratio of 80:20, and n-butyl mercaptan as the chain transfer agent with a mass fraction of 0.18%. After ultrasonic mixing, the solution is injected into the mold with a syringe as the cladding solution for the prefabricated rods, which are slowly heated to polymerize, and then the Teflon rope is drawn off to form a hollow prefabricated rod. Using vacuum

suction, the core solution is drawn into the air holes in the middle of the preform, consisting of MMA and BA in a volume ratio of 85:15, LPO and n-butyl mercaptan in the same ratio as the cladding, laser dye PM597 with a mass fraction of 0.14%, and magnetic scattering nanoparticles (NPs) with a mass fraction of 0.3%. The MGPOF is a structure of NPs packaged in a polymer matrix, which forms a solid structure when polymerization completed. Referring to the method in the literature [1], core-shell structured $\text{Fe}_3\text{O}_4@\text{SiO}_2$ magnetic NPs are fabricated. The preform is again polymerized by slowly heating them up and placed on the fiber drawing tower for melt stretching to obtain the MGPOF. The refractive index of the cladding is 1.4882 and the refractive index of the core is 1.4903. Figure S1(d) displays two microscope photographs of the MGPOF, illuminated by incandescent (above) and UV lamps (bottom), with a cladding diameter of $674 \mu\text{m}$ ($\pm 0.5 \mu\text{m}$) and a core diameter of $32 \mu\text{m}$ ($\pm 0.5 \mu\text{m}$). The length of the MGPOF is 5 cm, which is consistent with the diameter of the magnetic pole, so the whole MGPOF-RL is placed in the magnetic field environment.

I think more clarification about sample preparation and realization is needed and I suggest putting everything in one paragraph and not dispersed inside the main text, the methods and the supplementary sections. I find Figure 1 misleading and not relevant, maybe there is also some mistake in the caption (up and down instead of right and left).

Our response: We thank the reviewer for this important question. As mentioned above, we considerably modified Figure 1 and its caption. Moreover, we moved all information on the sample preparation to the Supplementary Material I. We also modified the text accordingly.

2. The main result of the manuscript is the tuning of the emission threshold by applying a magnetic field to the NPs. The light scattering is decreased by increasing field and the random laser gets less efficient and this is for sure a nice way for the modulation of RLs.

The authors demonstrate this by spectral (figure 2) and replica symmetry breaking (figure 3) analyses. The latter one has been proposed in many other works on different systems as a powerful tool to show RL action in glassy RLs with fluctuating emission.

Can the authors explain why in Figure 3 d-I, the RL results more efficient by increasing the magnetic field and thus by decreasing the scattering strength? Isn't it the opposite of their major claim?

A deep discussion is required on this point.

Our response: We thank the reviewer for raising this important question. This is actually a very keen observation. We are sorry for the lack of clarity in the original manuscript.

First, we should mention that the contents of the old Fig. 3 are now redistributed into the new Figs. 4 and 5 of the revised manuscript. In fact, in comparison with the old Fig. 3, the new Figs. 4 and 5 present more detailed results for a larger number of excitation energies, with further information on the behavior of Parisi overlap distribution $P(q)$ (Fig. 4), the maximum value $P(q)_{max}$ and associated standard deviation σ (Fig. 5). Regarding the referee's question, we should remark that the energy threshold also increases with the field, as seen in Fig. 2c of the revised manuscript. So, the increase of $P(q)_{max}$ with the magnetic field in Fig. 5 cannot be solely used to directly infer the efficiency of the RL, since the effect of the proximity of the threshold must be also taken into account in the analysis. To be more specific, consider, for example, the case with excitation energy 15 μJ in Fig. 4 (second column). Due to the dependence of the threshold on the field, we notice in Fig. 4g for 300 mT that the input energy 15 μJ is much closer to the threshold than in Fig. 4f, in which no field is applied. It is thus justified that $P(q)_{max}$ is larger in Fig. 4g than in Fig. 4f (with $P(q)$ more concentrated around the extrema $q = \pm 1$ in Fig. 4g), as actually observed. The same reasoning also applies for higher magnetic fields in Figs. 4h-4j with 15 μJ , and for other values of the excitation energy as well. This discussion has been added on page 18 of the revised manuscript.

3. To my opinion, spectral correlations (equation 5 and Figure 4) do not add any useful information. They confirm the results obtained by the spectral and RSB analysis.

Our response: We thank the reviewer for the comment. We have moved the original Equation 5 and Figure 4 to the Supplementary Material. Equation 5 and Figure 4 are

helpful to complement the discussion on the replica symmetry breaking analysis of the new Figure 4. The difference between the correlation coefficient and Parisi overlap parameter q is as follows: while the parameter q shows the spectrum-spectrum correlation summed over all wavelengths, the correlation coefficient infers how distinct wavelengths are correlated in the same spectrum. In particular, this combined analysis has appeared in a number of recent works, such as:

Xia J, Zhang X, Zhou K, Zhang L, Wang E, Du W, Ma J, Li S, Xie K, Yu B, Zhang J and Hu, Z, “Tunable replica symmetry breaking in random laser” *Nanophotonics* 12, 761-771 (2023).

Sarkar A, Bhaktha B S and Andreassen J, “Replica Symmetry Breaking in a Weakly Scattering Optofluidic Random Laser” *Scientific Reports* 10, 2628 (2020).

Coronel E, Das A, González I R, Gomes A S, Margulis W, Von Der Weid J P and Raposo E P, “Evaluation of Pearson correlation coefficient and Parisi parameter of replica symmetry breaking in a hybrid electronically addressable random fiber laser” *Optics Express* 29, 24422-24433 (2021).

4. Concluding, I think the many data reported in the manuscript, including supplementary material, are redundant and can be summarized in few significative ones. The only novel point of the manuscript is the ability to tune the RL emission and so the material structure and fabrication. The data analysis is not novel, the equations are well known and the theoretical part is not new, although useful.

I think after substantial revision in the material description, data presentation and organization, the manuscript can be published in a journal more specialized in optics/photonics.

Our response: We thank once again the reviewer for the quite nice suggestions and keen comments. We have made an effort to address every point raised and make all modifications suggested in the revised manuscript, in a sufficiently clear way for the benefit of a broad readership. Although the theoretical formalism is known, the application to the phenomena of photonic Hall effect and photonic magnetoresistance in RLs is novel, and certainly opens up further avenues to the understanding of such

effects in optical systems. Finally, we hope that, with these significant improvements, the referee can now consider our manuscript suitable for publication in *Nature Communications*.

Reviewer #3 (Remarks to the Author):

This manuscript presents an experimental study of a magnetic random laser under the influence of a strong applied magnetic field. Scatterers have diameters of tens of nanometers and are composed of magnetite with a shell of silicon dioxide [S1]. Optical gain is provided by laser dye doped inside an optical fiber core along with the nanoparticle scatterers, which are presumably randomly placed. Core diameter is 32 μm and cladding diameter 674 μm . Experiments are supported by scattering theory presented in the supplementary information, which shows the scattering coefficient decreasing with an increasing magnetic field as well as absorption and extinction increasing with an increasing magnetic field. Both effects should increase the lasing threshold, which is observed explicitly in Fig. 2(d). An increase of the threshold also means that the applied magnetic field is inversely related to the pump energy. Indeed, as an example in Fig. 4, increasing the magnetic field from 0 to 300 mT in (d) to (f) alters the heatmap to resemble that seen at lower pump energy in (a). However, as discussed in Refs. [1-3], the photonic Hall effect (PHE) and photonic magnetoresistance (PMR) depend specifically on the generation of a transverse light current. While it may be possible that such a current is generated in these experiments, there is not even a discussion of this phenomenon (for experimental observation see [2]). Moreover, replica symmetry breaking itself is not dependent on a transverse current. Therefore, while this is an interesting study on the effect of magnetism in random lasers, there is no conclusive evidence presented of either the PHE or PMR. We believe the PHE and

PMR could be speculatively discussed in the Conclusion of a revised manuscript, if the authors wish, but stating that this is the first observation of these phenomena in random lasers is misleading. There are also several questions that should be addressed, which are discussed below.

Our response: We are grateful to Reviewer #3 for investing time and expertise in reviewing our manuscript. We are glad about the positive evaluation and thank the referee for this critical comment, which we would like to address from several angles. We supplement the experiments described in the original manuscript with new results presented in the revised manuscript on the evidence of the transverse current that characterizes the photonic Hall effect, as also shown in the beginning of this response. Indeed, the results shown in the new Fig. 3(b) confirm the linear dependence of the magnetic transverse photocurrent generated by the PHE on the magnetic field strength in the RL system.

As for the photonic magnetoresistance (PMR), we believe that in the present work we have given conclusive evidence of it in RLs with, *e.g.*, the analysis of the emission intensity that decreases significantly when the magnetic field increases. In the revised manuscript, we highlight the novelty of these two effects in tuning the RL emission capacity, with significant revisions to the material description, data presentation, and organization.

Equation (1),

The contribution of light scattering to the experimental results is supported by the supplementary information even though it appears much smaller than contributions from absorption and extinction. However, just because light is being scattered differently with a magnetic field applied and altering the mean free path does not mean a transverse current has been established, which is a requirement for the PHE and PMR. Also, note that the quasimodes of random multiple-scattering systems are spatially random and light in the random fiber is already being deflected laterally (see Fig. 1 of [18]).

Our response: In the revised manuscript, we supplement the previous results with the

proof experiments for the photocurrent, as also shown in the beginning of this response. The quasimodes of random multiple-scattering systems are spatially random, but the disorder can be attenuated.

Figure 1 of Ref. [18] does not show the lateral deflection of the random scattering light, but the total reflection of the fiber waveguide.

Figure 1,

The diagrams here are confusing. From [S1], we believe the composition of nanoparticles is a silicon dioxide shell around a magnetite core. But in this figure, silicon dioxide is shown only on the bottom half of the scatterers. More importantly, below roughly 70 nm, magnetite nanoparticles have a single magnetic domain [Nature Sci Rep 7, 9894 (2017)]. This contradicts the authors' description of an external magnetic field aligning a distribution of magnetic moments in a single nanoparticle. Also, the placement of these scatterers in the fiber core, which is presumably random, is not discussed. There are a few types of disorder here that should be presented more clearly: (i) the amorphous nature of the silicon dioxide shell on each nanoparticle, (ii) the initial random orientation of the single-domain magnetic moment of each nanoparticle before a magnetic field is applied, and (iii) the spatially random placement of the nanoparticles in the optical fiber core. This figure requires significant revision.

Our response: We thank the reviewer for these suggestions. Nanoparticles are indeed core-shell structures. The upper and lower halves of the previous Fig. 1 are intended to depict the structural and field-dependent magnetic NP morphology. The spontaneous, noncorrelated spin disorder at the particle surface is strongly related to structural surface disorder. With an increasing magnetic field, the collinear magnetic volume overcomes the structurally coherent particle size.

The reference [Zácutná D, Nižňanský D, Barnsley L C, Babcock E, Salhi Z, Feoktystov A, Honecker D and Disch S, "Field Dependence of Magnetic Disorder in Nanoparticles", Physical Review X 10, 031019 (2020)] shows the significant increase of the magnetic moment of ferrite nanoparticles with an applied magnetic field, *e.g.*: "The performance

characteristics of magnetic nanoparticles toward application, e.g., in medicine and imaging or as sensors, are directly determined by their magnetization relaxation and total magnetic moment. In the commonly assumed picture, nanoparticles have a constant overall magnetic moment originating from the magnetization of the single-domain particle core surrounded by a surface region hosting spin disorder. In contrast, this work demonstrates the significant increase of the magnetic moment of ferrite nanoparticles with an applied magnetic field. At low magnetic field, the homogeneously magnetized particle core initially coincides in size with the structurally coherent grain of **12.8(2) nm diameter**, indicating a strong coupling between magnetic and structural disorder. Applied magnetic fields gradually polarize the uncorrelated, disordered surface spins, resulting in a magnetic volume more than 20% larger than the structurally coherent core. The intraparticle magnetic disorder energy increases sharply toward the defect-rich surface as established by the field dependence of the magnetization distribution. In consequence, these findings illustrate how the nanoparticle magnetization overcomes structural surface disorder. This new concept of intraparticle magnetization is deployable to other magnetic nanoparticle systems”

In the process of fiber fabrication, the magnetic nanoparticles are scattered in the fiber core disorderly, which agrees with the reviewer’s view.

We show the amorphous nature of each silica shell in the new Fig. 1. The purple arrows in the microscopic boxes at the bottom right indicate the structural disorder of the silica shell. We add the emphasis on the amorphous nature of the silica shell in the description of Fig. 1 in the main text.

Based on the reference mentioned above, magnetite nanoparticles cannot be simply described as having a single magnetic domain. So we do not plot in Fig. 1 the initial random direction of the single domain magnetic moment of each nanoparticle before applying the magnetic field, but we depict the spatially random placement of nanoparticles in the fiber core.

Lines 249-250, "The RL spectra are generated by averaging the 0th, 40th, 801st, 1201st, and 1601st spectra."

Why are these particular instances chosen? Just five spectra is statistically irrelevant and raises the question as to how data in Figs. 2(c), 2(d), and 3 are calculated. Hopefully all 2000 measurements were used in those calculations -- please clarify.

Our response: Since the RL emission is generated by the disordered multiple scattering of photons in the gain medium, the output intensity of each wavelength fluctuates randomly within a certain range. Clarifying the referee's point, in Fig. 3 of the previous manuscript (new Fig. 4 of the revised one), we have indeed used all 2000 measurements. Moreover, to improve the spectrum analysis, in the new Figs. 2(a-c) of the revised manuscript and Fig. S4 of the Supporting Information (see also below), we have now displayed results of data from all 2000 sets.

Modifications in the revised manuscript: [on page 13 in main text and page 6 in supplementary information]

The following new Figs. 2(a-c) of the revised manuscript and Fig. S4 of the Supporting Information have been considerably modified to show results from all 2000 measurements.

Figure 2. Field-dependent RL spectroscopy symbolizing PMR in RL. (a) RL spectra at the applied magnetic field intensity of 0, 50, 100, 200, and 300 mT at the pump energy of $25 \mu\text{J}$. (b) Variation of the maximum intensity of RL with the magnetic field at different pump energies. (c) Variation of the RL threshold with applied magnetic field intensity.

Figure S4. RL spectra at the applied magnetic field intensity of 0, 50, 100, 200, and 300 mT. Pump energy of 10 μJ for (a), 15 μJ for (b), 20 μJ for (c), 30 μJ for (d), 50 μJ for (e), and 100 μJ for (f). (g) Variation of the peak intensity of RL emission with pump energy. The inflection point of the fitted polyline corresponds to the threshold of the RL. Inset: The magnification of the inflection point.

Lines 258-259, "This feature can form a scattering closed loop when light is scattered between particles because the scattering disorder is strong enough."

Random laser modes formed by underlying quasimodes due to multiple scattering are more complex than can be described by just a "scattering closed loop" [e.g., Adv Opt Photon 3, 88 (2011)].

Our response: We thank the reviewer for raising this important point. We agree that the formation of RL modes is more complex than just "scattering closed loops". Whether it is a "closed scattering closed loop" picture, a "lucky photon" accumulating enough gain, or even a "longer-lived" mode with the coexistence of extended and localized modes, these different scenarios do not explain the full range of experimental observations reported in many works by several research groups, although they may be realized in some specific cases. In the work [Andreasen J, Asatryan A A, Botten L C,

Byrne M A, Cao H, Ge L, Labonté L, Sebbah P, Stone A D, Türeci H E and Vanneste C, *Adv. Opt. Photon.* 3, 88 (2011)], the authors show that the RL modes are associated with threshold lasing modes (TLMs), and constant-flux states are introduced to describe scattering medium for any scattering strength. Here we combine the claims in the above literature and attribute the fluctuations of the RL modes when the pump energy is varied to the RL modes being changed by the nonlinear modal interactions.

Modifications in the revised manuscript: [the first paragraph on page 12]

The nonlinear effects of saturation and mode competition will affect the performance of the pump⁴⁵. As the pump energy increases, the threshold of different modes will be reached, and new modes appear. General nonlinear theory is based on a self-consistent equation that determines how many modes exist in a fixed pump energy and the frequencies of these modes.

Figure 2(c),

Peak Intensity is shown in units of counts but the scale has a maximum of 1, which differs from what is shown in Figs. 2(a-c,f).

Our response: We thank the reviewer for the careful reading. In Fig. 2(c), the RL intensity at each pump energy with no field applied is considered as 1, and the degree of the RL intensity weakening with the increase of the magnetic field is calculated. The normalized values clearly show the degree of regulation of the RL intensity by the photonic magnetoresistance effect. Figures 2(a), (b), and (f) display the original values collected by the spectrometer. The following figure is the new Fig. 2(c).

Line 307,

What is meant by "time-induced weakening"? Bleaching of the laser dye?

Our response: Yes, the time-induced weakening means the bleaching of the laser dye.

After a considerable time of pumping, the life of the dye weakens.

Modifications in the revised manuscript: [the first paragraph on page 15]

The erratic RL intensity fluctuations rule out the attenuation of RL intensity by time-induced bleaching of the laser dye, which supports the modulating effect of the magnetic field on RL intensity.

Lines 322-324, "For very high pumping energies, the degree of disorder within the POF is greatly diminished and the RL modes oscillate coherently under nonlinear action."

This statement appears to conflict with Figs. 4(j)-4(l), where significant anti-correlations exist along with the absence of correlations between RL modes.

Our response: Thanks to the reviewer for pointing out this problem, on which we agree.

The appropriate sentence should be: "For very high pumping energies, the degree of disorder within the POF is greatly diminished and the replica-symmetric regime is recovered." It has been fixed in the revised manuscript.

Figure 3(f,i),

The values of $P(q)_{\max}$ do not seem to match the values observed in the $P(q)$ distributions, e.g., Fig. 3(d) shows $P(q) < 2e5$ but the value in Fig. 3(f) at $H = 0$ is above $3e5$. Such discrepancies are seen in the supplementary information as well.

Our response: We count the sum of the extreme values of the distribution $P(q)$ when the Parisi overlap parameter q_{ab} is positive or negative as

$$P(q)_{\max} = P(q)_{\max+} + P(q)_{\max-}, \quad (5)$$

meaning that $P(q)$ is actually given by the sum of the left and right side values. We apologize that we did not make that clear before. We have added a note on this issue in the Supplementary Information of the revised manuscript.

Figure 4(a-c),

What is the physical origin of the periodic, off-diagonal stripes? They are not random lasing modes, since the pump energy is below threshold. They cannot stem from the random quasimodes either, which are at random wavelengths (not regularly spaced).

Our response:

We first comment that we have moved the previous Fig.4 to the Supplementary Information (Fig.S6), as requested by Reviewer #2. We agree with Reviewer #3 that these wavelengths are not associated with random lasing modes or quasimodes. Indeed, these somewhat periodic off-diagonal stripes appear only below and around the threshold, possibly because the wavelengths of fluorescence emission are more stable with each other. The competition for gain in the fluorescent regime is at a comparable level and, as the system progressively enters the RL regime above threshold, the random character of the nonlinear couplings between field amplitudes becomes increasingly relevant. Interestingly, a similar phenomenon has been also reported below and around

the threshold in other RL and random fiber laser systems:

E. Coronel, A. Das, I. R. R. González, A. S. L. Gomes, W. Margulis, J. P. von der Weid, and E. P. Raposo, “Evaluation of Pearson correlation coefficient and Parisi parameter of replica symmetry breaking in a hybrid electronically addressable random fiber laser”, *Optics Express* 29, 24422-24433 (2021).

E. D. Coronel, M. L. da Silva-Neto, A. L. Moura, I. R. R. González, R. S. Pugina, E. G. Hilário, E. G. da Rocha, J. M. A. Caiut, A. S. L. Gomes, and E. P. Raposo, “Simultaneous evaluation of intermittency effects, replica symmetry breaking and modes dynamics correlations in a Nd:YAG random laser”, *Scientific Reports* 12, 1051 (2022).

E. D. Coronel, A. Das, M. L. da Silva-Neto, I. R. R. González, A. S. L. Gomes, and E. P. Raposo, “Statistical analysis of intensity fluctuations in the second harmonic of a multimode Nd:YAG laser through a modified Pearson correlation coefficient”, *Physical Review A* 106, 063515 (2022).

We thank the reviewer for raising this interesting question. We add the above description on page 11 in supplementary information.

Various errors in referencing other sections exist throughout the manuscript, e.g., should lines 290 and 436 refer to section VI of the supplementary information?

Our response: We thank the reviewer for the careful reading. We are sorry for this lapse. We have corrected this issue and reorganized the structure of the article.

Finally, we suggest including additional results at a higher applied magnetic field to match the hysteresis curve in the supplementary information. The optical response should qualitatively follow the hysteresis curve, i.e., at a high enough applied magnetic fields there should be no noticeable changes, meaning all magnetic effects have saturated.

Our response: Thanks to these important suggestions. We agree that the investigation under higher applied magnetic fields should be useful. The complete hysteresis loop is illustrated in the figure below, showing that the magnetization actually tends to saturate

from 3 T onwards. This magnetic field strength is not achievable with our current magnetic field generators. We hope in the future we will have better instrument to employ the PHE and PMR in RL to find the saturation point. Further, we hold opinion that this study is beyond the scope of this manuscript, which focuses on the first observation of PHE and PMR in RL systems.

Figure S1(b). Magnetic hysteresis loop at 300 K for $\text{Fe}_3\text{O}_4@\text{SiO}_2$ NPs, showing remnant magnetizations of 0.4 emu/g. The saturation magnetic field intensity is 50000 Oe, and the saturation magnetic induction intensity is 25.8 emu/g (Quantum Design, MPMS3).

Finally, we thank once again the reviewer for the quite nice suggestions and keen comments. We have made an effort to address every point raised and make all modifications suggested in the revised manuscript, in a sufficiently clear way for the benefit of a broad readership. We therefore hope that, with these significant improvements, the referee can now consider our manuscript suitable for publication in *Nature Communications*.

REVIEWER COMMENTS

Reviewer #1 (Remarks to the Author):

For me, the authors have addressed all my concerns and the paper has been improved deeply. Especially for Fig. 1, I believe the key idea of the paper can be understood clearly now. Thus, I can recommend this work for publication in Nature Communications in its present form.

Reviewer #2 (Remarks to the Author):

The manuscript has been improved after revision and my concerns are now addressed. I think it technically sounds and the results are important for the community working in the field of random lasers.

I do not see the wide impact appropriate for publications in Nature Communication.

Reviewer #3 (Remarks to the Author):

The authors have addressed most of our concerns sufficiently in this revised manuscript. Most importantly, evidence that the Photonic Hall Effect (PHE) may occur has been provided in their new Fig. 3. The approach is based on the seminal experimental observation of the PHE [2]. The net linear increase in the photonic current with respect to the applied magnetic field is convincing. Two issues remain that should be addressed prior to publication.

First, lateral scattering is certainly occurring in the absence of an applied magnetic field as light propagates through the fiber and is deflected laterally by the magnetic core-shell nanoparticles. This process is depicted by the red arrows in Fig. 1 of Ref. [18] and the yellow trajectory in Fig. 1 of this manuscript. Because of the random locations of the nanoparticles, which was clarified by the authors, there is no enforcement of spatial symmetry. Therefore, a "net zero" lateral deflection at any given point along the fiber should not be expected. Furthermore, there is no reason for there to be net zero lateral deflection at the output of the fiber either, even in the absence of an applied field. The simple physics here should be at least partially responsible for the non-zero intercept at zero field ($B=0$) in Fig. 3, where the photon current is greater than zero. Only the fact that this photon current depends on the applied field provides reasonable evidence that the PHE may occur.

Page 7, "The white arrows represent the collinear magnetic dipole spin in the magnetic core and the purple arrows represent the spin disorder of amorphous SiO₂ shell."

Second, there seems to be a fundamental misunderstanding of the magnetic nanoparticle behavior as indicated by the statement from the manuscript above. Silica has no relevant magnetic moment.

According to Ref. [37], the magnetic nanoparticle itself (just the magnetite) has a "defect-rich surface layer" filled with randomly oriented moments at zero applied field, which is several angstroms thick (see Table II [37]). This fact remains unclear in the revised manuscript and gets easily confused with the amorphous nature of the silica shell. The approach of [37] in applying a field to affect the thin, defect-rich magnetic surface layer is quite novel and different from the typical approach, e.g., [Nature Sci Rep 7, 9894 (2017)]. They had to explicitly show spatially resolved measurements to validate the notion that they could affect it. Since there is no direct evidence in this manuscript of spatial dependence of the magnetite on the applied field, the burden is on the authors to review [37] more thoroughly in the manuscript and provide a reason why their Fe₃O₄ nanoparticle should behave like the CoFe₂O₄ nanoparticle in [37]. Note that we agree the two compositions should behave similarly because of the surface atom distribution being different from the bulk, for which there are many references (e.g., [Int J Mol Sci 14, 15977 (2013)]), but this should be stated explicitly. Moreover, one reason this approach has only recently been done is because of the large magnetic field required for observation. It would be helpful to also relate the field strengths used in this manuscript to those in [37] (which are similar in the range 10-800 mT).

Response to the reviewers' comments

Reviewer #1 (Remarks to the Author):

“For me, the authors have addressed all my concerns and the paper has been improved deeply. Especially for Fig. 1, I believe the key idea of the paper can be understood clearly now. Thus, I can recommend this work for publication in Nature Communications in its present form.”

Our response: We are grateful to Reviewer #1 for investing time and expertise in reviewing our manuscript. We are glad about the positive evaluation on our work.

Reviewer #2 (Remarks to the Author):

“The manuscript has been improved after revision and my concerns are now addressed. I think it technically sounds and the results are important for the community working in the field of random lasers.

I do not see the wide impact appropriate for publications in Nature Communication.”

Our response: We are grateful to Reviewer #2 for investing time and expertise in reviewing our manuscript. We are also glad that Reviewer #2 has been satisfied with our answers (accompanied by the corresponding changes in the revised manuscript) to the quite keen points raised in the previous round.

Reviewer #2 also asks about the wideness of the impact of our work, on which we would like to elaborate as follows. Our work marks the inaugural experimental sighting of both the photonic Hall effect (PHE) and photonic magnetoresistance (PMR) in random lasers (RLs). This breakthrough has expanded the connection of microscopic magnetic particle scattering with macroscopic RL intensity fluctuations, proposing a means of regulating disorder in RL multiple scattering. The observation of PHE and PMR in RLs facilitates the exploration of the rich physical mechanisms underlying the

magneto-optical effects. We are confident about the significant impact of our work in various fields. In this sense, it nicely matches the scope of Nature Communications as being valuable for comprehensive communication between diverse and multidisciplinary audiences, such as physicists and material scientists. We also emphasize that RLs comprehend a highly influential area of research. As a typical representative of complex physical systems, RLs have recently attracted extensive interest from researchers in materials science, physics, statistics, biomedicine, optical communication, and other fields [see, e.g., Z. Xu et al., *Nano Lett.* 22, 172 (2022); B. Kumar et al., *Optica* 8, 1033 (2021); X. Shi et al., *Laser Photon. Rev.* 15, 7 (2021)]. From the perspective of application, for example, some RLs comprise nearly ideal representations of randomness and, for this reason, have attracted wide attention in the fields of encryption and time domain ghost imaging (also known as correlation imaging, which requires the intensity of the light source to fluctuate randomly) [A. S. L. Gomes et al., *Prog. Quant. Electron.* 78, 69 (2021)]. Moreover, from the perspective of basic principles of physics and mathematics, researchers try to find orderly rules in the phenomenon and mechanisms of RL disorder and regulate disorder by orderly means. We also remark that RL is a typical research area at the intersection of multiple disciplines. For instance, the following figure is a treemap chart of the research fields involved in searching the keyword “random laser” from Web of Science (<https://www.webofscience.com/wos/alldb/analyze-results/6dcd2d4a-fad4-49cc-a931-8cb73107744e-d943788a>). From this chart, we conclude that RLs actually have a wide impact in many research fields.

A summary of the above discussion has been added on page 3 of the revised manuscript (see also below in green).

Finally, we thank again Reviewer #2 for the quite nice suggestions and keen comments along the review process. We have made an effort to address every point raised, and make all modifications suggested in the revised manuscript, in a sufficiently clear way for the benefit of a broad readership. We therefore hope that, with these significant improvements, Reviewer #2 can now consider our manuscript suitable for publication in *Nature Communications*.

Modifications in the revised manuscript: (on page 3 of Introduction)

RLs are a prime example of a complex physical system and a key focal point for multidisciplinary research. RLs have attracted extensive research interest from researchers in materials science¹³, physics¹⁴, sensing¹⁵, biomedicine¹⁶, optical communication¹⁷, and other fields. From an application standpoint, the inherent randomness of RLs stands out as a rare embodiment of true randomness¹⁸, garnering significant interest in fields such as encryption¹⁷ and time-domain ghost imaging¹². Building on fundamental principles of physics and mathematics, researchers endeavor to discern underlying order within the phenomenon and mechanism of RLs disorder while seeking to regulate this disorder through systematic approaches¹⁹. As a result, more and more ordered regularities are found in RLs, while conventional lasers are introducing and benefiting from disordered elements.

Reviewer #3 (Remarks to the Author):

“The authors have addressed most of our concerns sufficiently in this revised manuscript. Most importantly, evidence that the Photonic Hall Effect (PHE) may occur has been provided in their new Fig. 3. The approach is based on the seminal experimental observation of the PHE [2]. The net linear increase in the photonic current with respect to the applied magnetic field is convincing. Two issues remain that should be addressed prior to publication.”

Our response: We are grateful to Reviewer #3 for investing time and expertise in reviewing our manuscript. We are also glad that Reviewer #3 has been satisfied with our answers (accompanied by the corresponding changes in the revised manuscript) to the quite keen points raised in the previous round.

“First, lateral scattering is certainly occurring in the absence of an applied magnetic field as light propagates through the fiber and is deflected laterally by the magnetic core-shell nanoparticles. This process is depicted by the red arrows in Fig. 1 of Ref. [18] and the yellow trajectory in Fig. 1 of this manuscript. Because of the random locations of the nanoparticles, which was clarified by the authors, there is no enforcement of spatial symmetry. Therefore, a "net zero" lateral deflection at any given point along the fiber should not be expected. Furthermore, there is no reason for there to be net zero lateral deflection at the output of the fiber either, even in the absence of an applied field. The simple physics here should be at least partially responsible for the non-zero intercept at zero field ($B=0$) in Fig. 3, where the photon current is greater than zero. Only the fact that this photon current depends on the applied field provides reasonable evidence that the PHE may occur.”

Our response: We thank the reviewer for raising this important question. This is actually a very keen observation. We agree that, in the absence of an applied magnetic field, lateral scattering certainly occurs when light propagates through the fiber and is laterally deflected by the magnetic core-shell nanoparticles. We also agree that there is no enforcement of spatial symmetry, and so we have revised Fig. 1 and related text

according to the comments by the reviewer (see below in green). Also, the intercept in Fig. 3(b) is indeed not zero, as noticed by the reviewer. This feature is now reinforced in the modified Fig. 1, in the green box in the upper right corner, where one can note that the scattered light in orange on the left has lateral deflection. We have also added a whole new paragraph on page 16 discussing, in the lines suggested above by the reviewer, the non-zero intercept at zero field ($B=0$) in Fig. 3 and the fact that a "net zero" lateral deflection of RL emission at any given point along the MGPOF should not be expected.

Modifications in the revised manuscript:

(Fig.1 on page 6)

(On page 16 of the main text:)

The results shown in Fig. 3(b) confirm the linear dependence of the magnetic transverse

photocurrent generated by the PHE on the magnetic field strength in a RL system. It is worth noting that the $\text{Fe}_3\text{O}_4@\text{SiO}_2$ magnetic nanoparticles are randomly distributed in the core of the MGPOF, so the MGPOF does not have perfect spatial symmetry. Therefore, a "net zero" lateral deflection of RL emission at any given point along the MGPOF should not be expected. When there is no applied magnetic field, $\Delta I_{\perp}/I_{\perp} \neq 0$. As the intensity of the applied magnetic field B increases, the magnetic transverse photocurrent increases. This linear dependence of the magnetic transverse photocurrent on the applied magnetic field shows the occurrence of PHE in the random laser. From Eq. (1) we conclude that the degree of PHE in the RL based on MGPOF is the slope of the red fitting curve $\eta = 1.099 \times 10^{-9}/\text{mT}$, which is the first evidence of PHE in RL systems.

“Page 7, "The white arrows represent the collinear magnetic dipole spin in the magnetic core and the purple arrows represent the spin disorder of amorphous SiO_2 shell.”

Second, there seems to be a fundamental misunderstanding of the magnetic nanoparticle behavior as indicated by the statement from the manuscript above. Silica has no relevant magnetic moment. According to Ref. [37], the magnetic nanoparticle itself (just the magnetite) has a "defect-rich surface layer" filled with randomly oriented moments at zero applied field, which is several angstroms thick (see Table II [37]). This fact remains unclear in the revised manuscript and gets easily confused with the amorphous nature of the silica shell. The approach of [37] in applying a field to affect the thin, defect-rich magnetic surface layer is quite novel and different from the typical approach, e.g., [Nature Sci Rep 7, 9894 (2017)]. They had to explicitly show spatially resolved measurements to validate the notion that they could affect it. Since there is no direct evidence in this manuscript of spatial dependence of the magnetite on the applied field, the burden is on the authors to review [37] more thoroughly in the manuscript and provide a reason why their Fe_3O_4 nanoparticle should behave like the CoFe_2O_4 nanoparticle in [37]. Note that we agree the two compositions should behave similarly because of the surface atom distribution being different from the bulk, for which there are many references (e.g., [Int J Mol Sci 14, 15977 (2013)]), but this should be stated explicitly. Moreover, one reason this approach has only recently been done is because

of the large magnetic field required for observation. It would be helpful to also relate the field strengths used in this manuscript to those in [37] (which are similar in the range 10-800 mT).”

Our response: This is actually a quite keen point raised by the reviewer. We agree that the silica shell has no magnetic moments, and we thank the reviewer for pointing this out. According to Ref. [37] the magnetic nanoparticles themselves have a “defect-rich surface layer” in zero magnetic field, which is filled with randomly oriented moments and has a thickness of several angstroms. We have thus corrected this misconception in the revised manuscript by now stating on page 8 that “The white arrows represent the collinear magnetic dipole spin in the magnetic core and the purple arrows represent the spin disorder of Fe₃O₄ surface region.” (see also below in green).

Prior to the report in [37], all studies on the spin structure of magnetic nanoparticles relied on a static image of a constant, field-independent nanoparticle torque, because the test method in [37] required a large magnetic field to observe that the size of the magnetized core changed significantly with the applied magnetic field. We now carefully review the results and statements of Ref. [37] in the new pieces of text added on pages 7 and 8 of the revised manuscript (see also below).

The unique nature of magnetic NPs stems from the fact that these nanoscale magnets are different from bulk materials due to their high surface to volume ratio. We believe that the core-shell structure ferric oxide nanoparticle sphere used in this paper and the CoFe₂O₄ nanoparticle sphere in [37] surrounded by the oleic acid ligand layer behave similarly. That is, the size of the magnetized core increases with the applied magnetic field because they have similar surface atom distribution and similar magnetic resonance lines in the range 0-800 mT. The two particles should have similar behavior according to the following works: A. G. Kolhatkar et al., *Int. J. Mol. Sci.* 14, 15977 (2013); A. H. Lu et al., *Angew. Chem. Int. Ed.* 46, 1222 (2007); S. Singamaneni et al., *J. Mater. Chem.* 21, 16819 (2011)]. We have added these three references as Refs. [38-40], and also introduced the above explanation in the revised manuscript (see also below).

Modifications in the revised manuscript:

(On page 7 of the main text:)

From the microscopic disordered multiple scattering during RL generation, we note that contrary to the commonly assumed view that nanoparticles have a constant global magnetic moment, in fact, the applied magnetic field can significantly increase the magnetic moment of ferrite nanoparticles. The magnetization volume of nanoparticles is closely related to the surface disorder of their structure, and when a magnetic field is applied, the disordered surface spin is gradually polarized, resulting in an increase of more than 20% in the magnetic volume [37]. In Ref. [37], the magnetic scattering amplitude of small angle neutron scattering (SANS) is used to determine the morphology of magnetic nanoparticles. It is found that the magnetic core radius r_{mag} increases with the increase of the external magnetic field strength. The variation of the severe magnetic scattering amplitude with the applied magnetic field is simulated based on the micromagnetic theory. The simulation results show that the fluctuation of magnetic parameters, that is, the contribution of magnetocrystalline anisotropy and magnetostriction, is the most likely source of the variation of magnetic radius with magnetic field. The magnetic volume and the corresponding magnetic field energy increase with the increase of the applied magnetic field strength are obtained by calculating the Zeeman free energy. The core-shell $\text{Fe}_3\text{O}_4@\text{SiO}_2$ NPs used in this work and the CoFe_2O_4 nanoparticle sphere in [37] surrounded by the oleic acid ligand layer behave similarly. That is, the size of the magnetized core increases with the applied magnetic field because they have similar high surface-to-volume ratios, similar surface atom distribution, and similar magnetic resonance lines at 0-800 mT³⁸⁻⁴⁰.

(On page 8 of the main text:)

The deep pink squares show the structurally coherent grain of the Fe_3O_4 core, and the light pink dots show the structural disorder of the Fe_3O_4 surface region. Note that the SiO_2 shell is not shown in the morphology of ferrite NPs. The silica shell on each NPs has its amorphous nature. The white arrows represent the collinear magnetic dipole spin in the magnetic core and the purple arrows represent the spin disorder of Fe_3O_4 surface

region. The overall magnetic moment of the core-shell structure $\text{Fe}_3\text{O}_4@\text{SiO}_2$ magnetic NPs increases with the magnetic field.

Finally, we thank again Reviewer #3 for the quite nice suggestions and keen comments. We have made an effort to address every point raised, and make all modifications suggested in the revised manuscript, in a sufficiently clear way for the benefit of a broad readership. We therefore hope that, with these significant improvements, Reviewer #3 can now consider our manuscript suitable for publication in *Nature Communications*.